



# Impact of North Brazil Current rings on air-sea CO2 flux variability in winter 2020

Léa Olivier[1], Jacqueline Boutin[1], Gilles Reverdin[1], Nathalie Lefèvre[1], Peter Landschützer[2], Sabrina Speich[3], Johannes Karstensen[4], Markus Ritschel[2] and Rik Wanninkhof[5]

[1] LOCEAN-IPSL, Sorbonne Université-CNRS-IRD-MNHN, Paris, France
[2] Max Planck Institute for Meteorology, Hamburg, Germany
[3] Laboratoire de Météorologie Dynamique, ENS-Ecole Polytechnique-CNRS-Sorbonne Université, Paris, France
[4] GEOMAR Helmholtz Centre for Ocean Research, Kiel, Germany
[5] Atlantic Oceanographic & Meteorological Laboratory of NOAA, Miami, USA

*Correspondence to*: Léa Olivier (lea.olivier@locean.ipsl.fr)

**Abstract.** The North Brazil Current (NBC) flows northward across the Equator, passes the mouth of the Amazon River, and forms large oceanic eddies near 8°N. We investigate the processes driving the variability of air-sea $CO_2$ fluxes at different scales in early 2020 in the region [50°W-59°W – 5°N-16°N]. This region is a pathway between the equatorial and North Atlantic Ocean and was surveyed during the EUREC[4]A-OA/ATOMIC campaign. In-situ surface fugacity of $CO_2$ ($fCO_2$), salinity and temperature combined with maps of satellite salinity, chlorophyll-a and temperature highlight contrasting properties in the region. In February 2020, the area is a $CO_2$ sink (-1.7 TgC.month$^{-1}$), previously underestimated by a factor 10. The NBC rings transport saline and high $fCO_2$ water indicative of their equatorial origins and are a small source of $CO_2$ at regional scale. Their main impact on the variability of biogeochemical parameters is through the filaments they entrain into the open ocean. During the campaign, a nutrient rich freshwater plume from the Amazon River is entrained from the shelf up to 12°N and caused a phytoplankton bloom leading to a significant carbon drawdown (~20 % of the total sink). On the other hand, saltier filaments of shelf water rich in detrital material act as strong local sources of $CO_2$. Spatial distribution of $fCO_2$ is therefore strongly influenced by ocean dynamics south of 12°N. The less variable North Atlantic subtropical water extends from Barbados northward. They represent ~60 % of the total sink due to their lower temperature associated with winter cooling and strong winds.

## 1 Introduction

The North Brazil Current is one of the dominant features of the tropical Atlantic circulation. In a region dominated by zonal jets, it flows northward along the coast of South America and separates from the coast around 6-8°N. It seasonally retroflects to feed the North Equatorial Counter Current and thus close the equatorial wind-driven gyre. This retroflection occasionally



pinches off some of the world's largest eddies, the North Brazil Current rings (NBC rings, Johns et al., 1990; Richardson et al., 1994).

After their separation from the NBC retroflection region, the rings travel north-westward toward the Caribbean in a course parallel to the coast of South America. These eddies have been extensively studied using modelling and both, in-situ (e.g. 1998-2001 NBC Ring experiment, Wilson et al., 2002) and satellite observations. They have a mean radius of 200 km and their diameter can exceed 450 km. Vertically, some of them extend down to more than 1000 m (Fratantoni & Glickson, 2002; Fratantoni & Richardson, 2006; Johns et al., 2003). Different family of rings exists, and most of the anticyclonic eddies

detectable by altimetry are rather shallow, extending from the surface to 200-300 m (Garraffo et al., 2003; Wilson et al., 2002). They swirl clockwise and travel with an average north-westward translation speed of 8-15 km d$^{-1}$ (Johns et al., 1990; Mélice & Arnault, 2017). When they reach the Lesser Antilles, they start to coalesce and disintegrate, partly due to interactions with the topography (Fratantoni & Richardson, 2006; Jochumsen et al., 2010). There is substantial variability in the number of rings shed per year, ranging from 5 (Aroucha et al., 2020; Fratantoni & Glickson, 2002; Mélice & Arnault, 2017, Goni & Johns,

2001) to 9 (Johns et al., 2003). NBC rings play a crucial role in the interhemispheric transport of salt and heat in the Atlantic Ocean and are an important part of the meridional overturning circulation (Johns et al., 2003). The NBC rings disrupt an already complex region located in the vicinity of the Amazon River mouth and at the transition between equatorial and subtropical waters.

While most of the studies on rings focused on their physical properties, little is known about their biogeochemical properties and how they affect the air-sea $CO_2$ flux of the western tropical Atlantic. The global ocean acts as an atmospheric $CO_2$ sink, taking up 23% of total anthropogenic $CO_2$ emissions (Friedlingstein et al., 2020) and causing ocean acidification (IPCC, 2019; 2021). The equatorial Atlantic Ocean is the second-largest source of $CO_2$ to the atmosphere after the equatorial Pacific (Landschützer et al., 2014; Takahashi et al., 2009). The concentration of atmospheric $CO_2$ is continuously increasing due to

human activities (IPCC, 2019; 2021), and characterizing the role of the ocean in mitigating climate change through $CO_2$ uptake is thus a key investigation. Previous works in this region examined the influence of the equatorial upwelling and of the Amazon plume on the $CO_2$ flux. $CO_2$-rich equatorial waters, originating from the equatorial upwelling (Andrié et al., 1986) strongly contrast with the $CO_2$ undersaturated Amazon River plume waters. The magnitude of the Amazon River discharge is unique in the global ocean. It represents as much freshwater as the next 7 largest rivers in the world combined and contributes to

almost 20% of global river freshwater input to the ocean (Dai & Trenberth, 2002). It therefore strongly impacts the physical, biogeochemical and biological properties of the coastal and the open ocean. Often overlooked, the Amazon River plume is an atmospheric $CO_2$ sink of global importance (Ibánhez et al., 2016). The plume carries water rich in silicate, nitrogen and phosphate into the tropical oceanic waters that are strongly depleted in nutrients. As water mixes and turbidity decreases, the primary producer's growth and associated biological drawdown are stimulated (Chen et al., 2012). Nitrogen is rapidly

consumed, and nitrogen fixation by diazotrophs becomes the main pathway of carbon sequestration in the plume



(Subramaniam et al., 2008). This strong carbon drawdown leads to a significant sink of atmospheric $CO_2$ (Körtzinger, 2003; Lefèvre et al., 2010). Not taking into account the Amazon plume would result in overestimating the tropical Atlantic air-sea $CO_2$ flux by 10% (Ibánhez et al., 2016).

The Amazon River's discharge reaches a minimum in December and progressively increases from January onwards. The plume extension is minimum in that season (Fournier et al., 2015) and as a result, it is the season of maximum salinity in the northwestern tropical Atlantic. The difference between the fugacity of $CO_2$ ($fCO_2$) in the surface ocean and in the atmosphere ($\Delta fCO_2$) climatology over 1998-2015 divides the north-western tropical Atlantic into two regions (Figure 1a). The Amazon outflow region is particularly hard to reconstruct due to its strong variability and a severe lack of data. Waters located in the

southeasternmost part of the domain act as a strong source of $CO_2$ to the atmosphere. The source progressively transitions into a sink north of 10°N as waters get colder linked to the seasonal winter cooling. This situation is typical of a transition zone between equatorial and subtropical waters in winter (Landschützer et al., 2020, Figure 1a). The freshwater of the Amazon remains mainly confined to the continental shelf due to winds perpendicular to the coast as they travel northwestward into the Caribbean Sea (Coles et al., 2013). However, it has recently been documented that off-shore freshwater transport is often

present in February and significantly alters the physical properties of the region (Reverdin et al., 2021). This is partly due to the interaction of the NBC rings with the Amazon plume (Figure 1c). The ocean color signature of the Amazon (Muller-Karger et al., 1988) has been used as a tracer to delineate the rings, and better understand their generation, evolution and characteristics (Johns et al., 1990; Fratantoni & Glickson, 2002). For example, Figure 1b shows two filaments stirred by two large rings. The Amazon River also influences the surface temperature and salinity of the rings. They are considered warm-core rings but have

a warm SST anomaly in the first half of the year, and a cold one in the second half, because the anomaly is relative to the regional SST, with an extensive warm pool in late summer and autumn (Ffield, 2005). Their signature in salinity is therefore plume-dependent as well. Ffield (2005) reported that 3 out of 4 rings were surrounded by lower salinity water. This is also highlighted in February 2020 where an NBC ring stirs a plume of fresher water rich in chlorophyll-a toward the open ocean (Figure 1b,c). Salinity and chlorophyll-a are therefore critical to understand the surface physical and biogeochemical properties

of the region, as well as the air-sea fluxes of $CO_2$ (Figure 1).

The northwestern tropical Atlantic is a dynamically active region, containing eddies several hundred kilometers in diameter and connected to the world's largest river. There are surprisingly few biogeochemical observations available for winter months during low outflow from the Amazon River. Few tropical Atlantic measurements of biogeochemical tracers are available with

one transect in winter in the Surface Ocean CO2 Atlas (SOCAT, Bakker et al., 2016) database south of 10°N crossing the region. This scarcity is a major impediment in understanding the biological and physical processes underlying the oceanic carbon and nutrient cycles in the region. Satellite surface chlorophyll-a and salinity show a strong spatial structure, with eddies, filaments, and a fresh water plume (Figure 1 b,c). In this study, we take advantage of the physical and biogeochemical data





collected during the EUREC⁴A-OA/ATOMIC experiment in January-February 2020, combined with satellite data, to
understand how the NBC rings and their related structures impact the air-sea fluxes of $CO_2$ in winter.

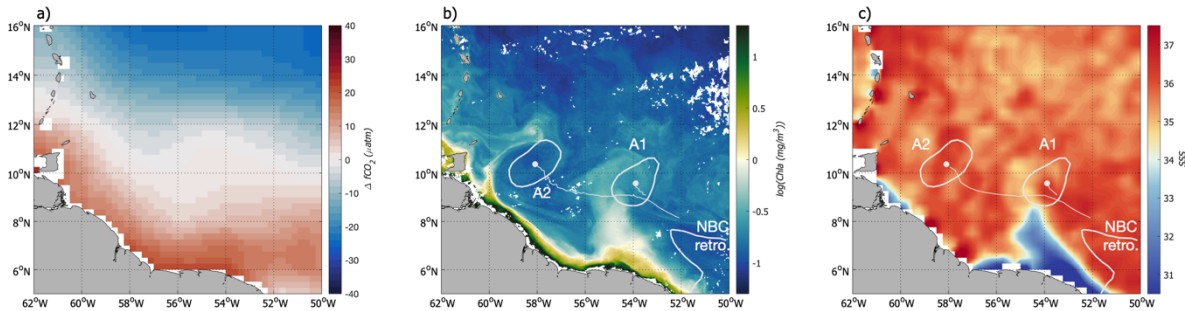

**Figure 1: ΔfCO2 February 1998-2015 climatology (Landschützer et al., 2020). b) Chlorophyll-a and c) SSS on Feb. 6th with the contours of NBC rings A1 and A2 defined by the TOEddies algorithm (Laxenaire et al. 2018) from concurrent satellite Absolute**
**Dynamic Topography (ADT) field, their center and trajectory. The 0.51 m contour of satellite derived ADT represents the NBC retroflection.**

The paper is organized as follows. We present the in-situ observational data from the EUREC⁴A-OA/ATOMIC experiment as
well as the satellite data in section 2. We identify the water masses observed in the region, their physical and biogeochemical properties and estimate the $CO_2$ fluxes at regional scale using empirical relationships in section 3. We compare the results with climatologies of $CO_2$ fluxes to evaluate the added knowledge brought by the intensive surveys of February 2020 and discuss the interannual variability in section 4 followed by the conclusion.

## 2 Data and Methods

### 2.1 In-situ data

The EUREC⁴A-OA (Elucidating the Role of Clouds-Circulation Coupling in Climate Ocean-Atmosphere) / ATOMIC (Atlantic Tradewind Ocean Mesoscale Interaction Campaign, Stevens et al., 2021) campaign took place in January and February 2020 and involved research vessels (RV) from France (RV Atalante, Speich & The Embarked Science Team, 2021), Germany (RV Maria S. Merian, hereby designated as Merian, Karstensen et al., 2020 and RV Meteor, not considered in this
study since no $CO_2$ measurements were taken onboard), and the United States (RV Ronald H. Brown, hereby designated as Ron Brown, Quinn et al. 2021). These cruises provided numerous in-situ measurements and, in this study, we will focus on the continuous near surface measurement of temperature, salinity, fCO₂ and winds.



Temperature, salinity from thermosalinographs (TSG), as well as $fCO_2$, were measured from water pumped ~5 m below the
surface (Figure 2 a,b,c, $\Delta fCO_2$ from measured $fCO_2$). For each ship, the resulting $CO_2$ data is corrected (Lefèvre et al., 2010;
Pierrot et al., 2009) from the temperature difference between the water at the ship's water intake and the one analyzed by the
instrument. On RV Atalante, underway oceanic and atmospheric $fCO_2$ were detected by infrared detection using a Licor 7000
(Olivier et al., 2020). The $fCO_2$ system was the same as in Lefèvre et al. (2010). It uses an air–sea equilibrator described by
Poisson et al. (1993). Seawater from the TSG pumping circuit circulates in the equilibrator at a rate of 2 L.min$^{-1}$. A closed-
loop of about 100 ml of air flows through the equilibrator designed to avoid bubbles at the air–sea interface. To minimize
temperature corrections, the equilibrator is thermostated with the same seawater as the one used for $CO_2$ measurements. The
temperature difference between the equilibrator and the sea was on the order of 0.5 °C.

Furthermore, 138 samples for dissolved inorganic carbon (DIC) and total alkalinity (TA) analysis were collected onboard RV
Atalante as well as inorganic nutrients (silicate, phosphate, nitrate and nitrite).

An OceanPackCUBE ferrybox system from SubCtech was installed on the RV Merian measuring continuously the oceanic
$fCO_2$. Water is pumped at a rate of ~7 l/min through a debubbler unit subsequently followed by a SeaBird SBE 45
thermosalinograph before it circulates along a membrane through which $CO_2$ diffuses. On the other side of the membrane, the
air-loop is circulated at a rate of 0.5 l/min through a Li-COR LI840 non-dispersive infrared gas analyzer (e.g., Arruda et al.,
2020). On RV Ron Brown the General Oceanic Inc. 8500 $pCO_2$ instrument follows a similar methodology to the underway
$fCO_2$ system deployed on the Atalante and is detailed in Pierrot et al. (2009).

Wind measured by the weather stations of each research vessel is adjusted using a logarithmic wind velocity profile to represent
the wind at 10 m height above sea level which is used to compute the air-sea $CO_2$ fluxes.

An intercomparison of the $fCO_2$ measured by the RVs Atalante and Merian is attempted when the ships were located at a
distance inferior to 5 km (Figure 2d). On average, the $fCO_2$ measured by the RV Merian is 6.4 µatm higher than the one
measured on the RV Atalante, with a standard deviation of 4.8 µatm. The RVs Merian and Ron Brown crossed the same water
mass at 13-14°N/57°W on February 12$^{th}$. On average, the RV Merian $fCO_2$ is 6 µatm higher than the RV Ron Brown $fCO_2$. In
part we link these differences to the slower response time of the membrane system, however differences also lie within the
uncertainties of the $fCO_2$ observing systems (~5µatm for the membrane system installed on the RV Merian (see Arruda et al.,
2020) and ~2 µatm for the equilibrator systems installed on the RVs Atalante and Ron Brown). The region where RV Merian
and RV Atalante were very close is very variable (standard deviation of 20 µatm), and the RV Merian and RV Ron Brown
were never in the same place at the same time, so that the observed differences could also be due to the natural variability of
$fCO_2$ sampled differently by the various ships. Hence, we did not apply any correction and we checked that the effect of a 6
µatm systematic bias on the RV Merian $fCO_2$ has a minor effect on our resulting interpolations. It would lead to less than 2
µatm difference on our mean interpolated $fCO_2$ and less than 0.1 mmol.m$^{-2}$.day$^{-1}$ on the mean derived air-sea $CO_2$ flux.





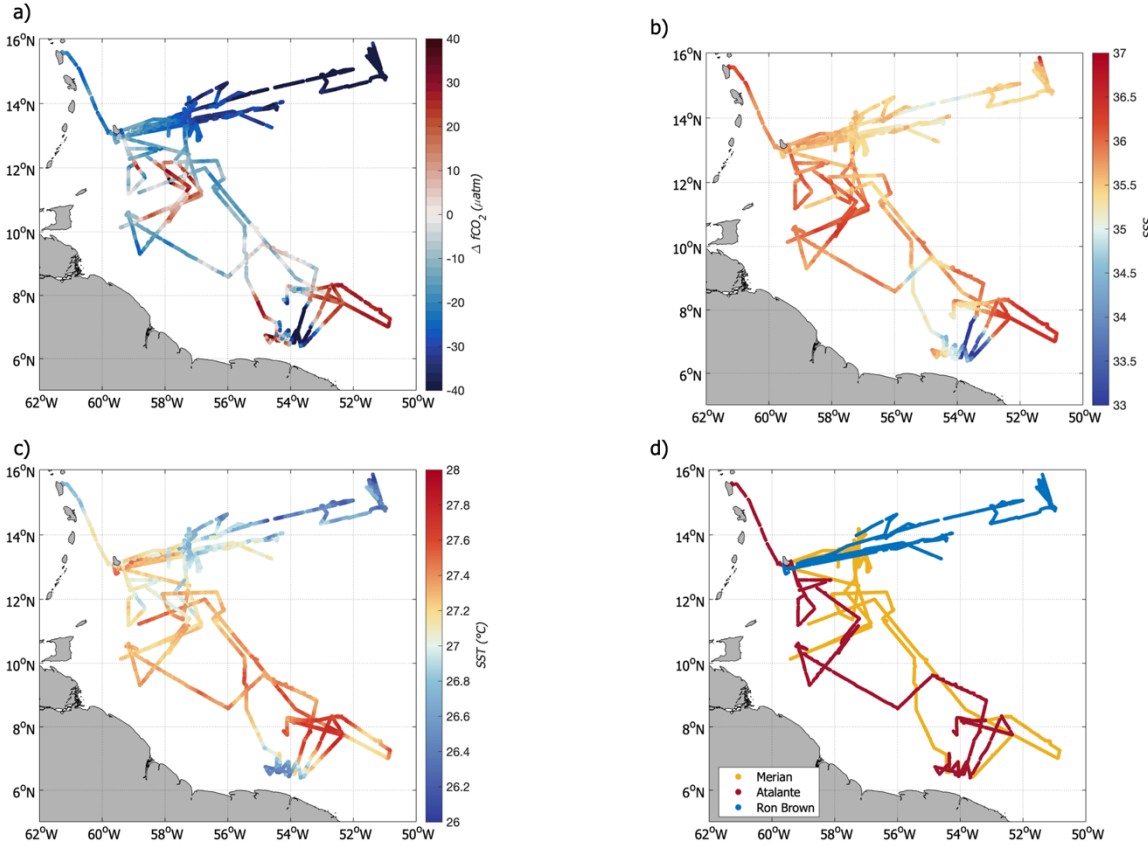

Figure 2: In-situ measurements of surface (a) ΔfCO2, (b) salinity and (c) temperature. The Merian, Atalante and Ron Brown ship tracks are presented in (d).

## 2.2 Satellite and atmospheric reanalyzes data

Daily satellite maps of chlorophyll-a (Chl*a*), sea surface temperature (SST), as well as absolute dynamic topography (ADT) and sea surface salinity (SSS) are used in this study.

The salinity maps are a blend of the Soil Moisture Ocean Salinity (SMOS, Jan. 2010-present), and Soil Moisture Active Passive (SMAP, Apr. 2015-present) measurements developed by Reverdin et al. (2021) and optimized for the northwestern tropical

Atlantic in February 2020 (Figure 1c). The European SMOS and US SMAP missions observe the sea surface by L-band radiometry from sun-synchronous polar-orbiting satellites (Entekhabi et al., 2010; Font et al., 2009; Kerr et al., 2010; Piepmeier et al., 2017). Combining 6 a.m. and 6 p.m. measurements of both missions provides an almost complete coverage each day.





When the coverage was not complete over our region, the 6 a.m. track of the following day was also included. This daily field is available for the first 20 days of February, and leaves out only 2 days without sufficient coverage to retrieve salinity data. It

has a spatial resolution close to 70 km, and an uncertainty on the order of 0.5 pss. The product, its uncertainties and the comparison between the TSG salinity and the SSS product are detailed in Reverdin et al. (2021).

Daily Chl$a$ concentration maps and SST maps are produced by CLS (Stum et al., 2016) on a spatial grid of 0.02°. The Chl$a$ concentration maps are composites built from VIIRS (on Suomi-NPP and NOAA-20 US platforms) and OLCI (on Sentinel

3A and 3B Copernicus European platforms) satellite sensors. The SST product is a 1-day average of 4 infrared radiometer satellite data. Both datasets are sensitive to the cloud cover, but during our period of interest, they are usually without many gaps except at the end of February. Comparison between the TSG SST and the satellite SST product are detailed in the RV Atalante cruise report (Speich & The Embarked Science Team, 2021).

Daily ADT maps at a ¼° resolution combine data from all satellites available for the period 1993 to present. From these ADT maps, the TOEddies algorithm, developed by Laxenaire et al. (2018), identifies eddies and their trajectories (Figure 1). The eddy detection is based on the closed contours of ADT, as well as the maximum geostrophic velocity associated to the eddy.

In order to compute the air-sea $CO_2$ fluxes, the European Centre for Medium-Range Weather Forecasts (ECMWF) Reanalysis

v5 (ERA5) hourly wind speed and mean sea level pressure, $P_{atm}$, is used. ERA5 covers the period from January 1950 to present, and provides hourly data on a 30 km grid. In addition, the monthly wind speed and SST fields over the period 1998-2015 are used, and a climatology over this period is computed. The wind speed in the region in winter is on average between 6 and 8 m/s and its variability is low.

We compare the EUREC[4]A-OA/ATOMIC observations with the observation-based $CO_2$ partial pressure (pCO$_2$) climatology developed by Landschützer et al., (2020), created using a 2-step neural network method (Landschützer et al., 2016) and combining open and coastal ocean datasets. The associated $\Delta pCO_2$ and air-sea $CO_2$ flux monthly field climatologies over the 1998-2015 period are computed using the ERA5 climatological wind, SST and $P_{atm}$ fields as well as the atmospheric $CO_2$ from the Ragged Point, Barbados station.

**2.3 Methods**

**2.3.1 Air-sea CO2 flux**

We compute the air-sea flux (F; mmol.m$^{-2}$.day$^{-1}$) as:

$$F = k \cdot K_0 \cdot (fCO_2 - fCO_{2atm}) \tag{2}$$



Where $K_0$ is the solubility of $CO_2$ is seawater, expressed as a function of SSS and SST by Weiss (1974); $fCO_{2atm}$ is the

atmospheric $CO_2$ fugacity; and k is the gas transfer velocity. k is calculated following the relation from Wanninkhof (2014):

$$k = 0.251 \cdot < U^2 > \cdot (Sc/660)^{-0.5} \qquad (3)$$

where Sc is the Schmidt number and U is the wind speed at 10 m above sea level measured by each ships weather station or

derived from ERA5 wind speed. The measured winds from the ships are used to compute the along-track flux, while the ERA5

wind speed is used for the satellite-based analysis and the air-sea $CO_2$ flux climatology.

In order to compute $fCO_{2atm}$ over that period, we first derived the saturation vapor pressure ($P_{H2O}$) then the atmospheric $pCO_2$

using the monthly averaged $CO_2$ mole fraction ($xCO_{2atm}$) measured at the NOAA/Earth System Research Laboratory (ESRL)

station in Ragged Point, Barbados (13.17°N, 59.43°W):

$$fCO_{2atm} = xCO_{2atm} \cdot (P_{atm} - P_{H20}) \cdot C_f \qquad (1)$$

Where $P_{H2O}$ is the saturation vapor pressure computed from SSS and SST, and $C_f$ is the fugacity coefficient, function of the

atmospheric pressure and SST (Weiss, 1974).

### 2.3.2 Reconstruction of $fCO_2$ from satellite maps

We observed that in the northwestern tropical Atlantic in winter the in situ $fCO_2$ strongly depends on SST, SSS and surface

Chl*a* (Figure 3). There is a strong positive dependence of $fCO_2$ on SSS, with low $fCO_2$ for low SSS (Figure 3a). Across the

whole EUREC⁴A-OA/ATOMIC region, SST did not vary much (mean SST of 27°C, and standard deviation of 0.5°C), but

warmer waters present higher $fCO_2$. The dependence on Chl*a* allows for the discrimination of water masses with the same

surface TS properties but not the same $fCO_2$. $fCO_2$ is not only influenced by ocean dynamics and chemistry, but also by marine

biology. The biological carbon pump is one of the major components of the oceanic and global carbon cycles, as the

photosynthetic production of organic carbon by marine phytoplankton accounts for about half of the carbon fixation associated

with global primary production (Arrigo, 2007; Behrenfeld et al., 2006; Field et al., 1998). Satellite-based Chl*a* is hard to

discriminate from detrital material in areas were both are present and have the same wavelength. Figure 3 shows that waters

with a SST of 26.5°C and SSS between 35-36 can either be rich in Chl*a* and have a high $fCO_2$ or low in Chl*a* and have a low

$fCO_2$. Waters rich in detrital material tends to limit the phytoplankton growth and microbial respiration of riverine material on

the continental shelf likely dominates (Aller & Blair, 2006; Medeiros et al., 2015; Mu et al., 2021). This relationship is analyzed

in more detail for the different water masses identified in the region in parts 3.2. and 3.3. Chl*a* was not measured onboard,

thus, we use satellite surface Chl*a* co-located along the ship track.

Our approach is to derive from this large dataset a relationship linking $fCO_2$ to SST, SSS and Chl*a* in order to provide maps

of $fCO_2$ based on the satellite maps of SST, SSS and Chl*a*. Even considering the extend of the EUREC⁴A-OA atomic cruise,

the dataset is still sparse, and cannot fully represent the small-scale variability it highlights. In order to understand the fluxes

at regional scale the need for a good spatial resolution arises. For that, the surface T-S-Chl*a* diagram computed from the ship



measurement (and collocated satellite Chl*a*) is interpolated using a linear 3D interpolation on a grid of SST, SSS and Chl*a*. The grid has a resolution of 0.01°C in SST, 0.1 in SSS, and 0.01 in log(Chl*a* (mg/m³)). Using a 3D linear interpolation to mapping the fCO₂ data over a grid is a simple yet effective solution for a dataset that is still relatively sparse. The method is

presented in more details in the supplementary materials (Text S1). Using a linear fit prevents from oscillations between two data points, and yields good results. Along the ship track, the standard deviation between the measured and reconstructed fCO₂ is of 4 µatm. To each triplet of surface T, S, and log(Chl*a*) in the range of the values measured by the ship is therefore associated a value of fCO₂ based on the 3D linear interpolation of the in-situ values. In order to cover the whole range of T-S-log(Chl*a)* present in the region, we extrapolate to lower temperatures and lower salinities than the ones measured by the ship. In order to

do so, we add 4 points to the T-S-Chl*a* diagram at lower salinities and lower temperatures based on previous knowledge of the region. For the low salinity domain (SSS < 30), fCO₂ is strongly dominated by salinity and the influence of temperature is weak (Lefèvre et al., 2010). The SSS-fCO₂ relation developed by Lefèvre (2010) is in good agreement with the SSS-fCO₂ relationship computed from this study data (supplementary Figure S1) in the common range, we therefore use it to compute fCO₂ at a salinity of 26 (fCO₂(S = 26) = 251.4 µatm). The lower temperature is mostly located in the northern part of the

domain, that is the least variable and where the variations of fCO₂ are dominated by the ones in temperature. From this dataset

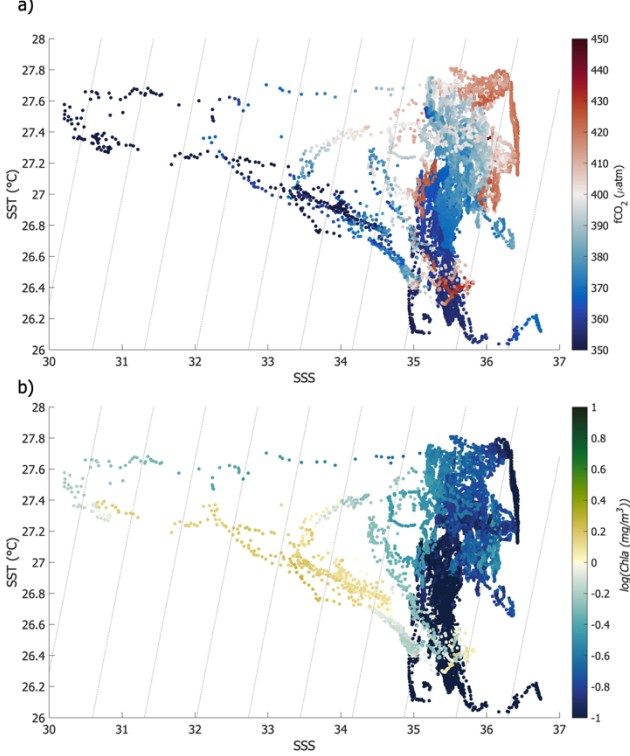

**Figure 3: Surface TS diagram, color-coded with fCO₂ (a) and Chl*a* (b).**



we compute a variation of 15 μatm/°C, which is consistent with the 4.23% °C$^{-1}$ expected variation of fCO$_2$ with temperature due to the temperature sensitivity of the carbonate dissociation constants and CO$_2$ solubility (Takahashi et al., 2002; Wanninkhof et al., 1999). We use this dependency to compute the fCO$_2$ at a temperature of 24°C to extend cover the whole range of temperature in the region.


We combine the interpolated fCO$_2$ with satellite maps of SST, SSS and Chl*a* to obtain daily high-resolution maps of fCO$_2$. Some days, either the presence of clouds altering the Chl*a* and SST or the lack of salinity coverage prevent the retrieval of fCO$_2$. In order to limit the error on fCO$_2$, we only keep 9 days out of the 20 first days of February (2, 4, 6, 7, 9, 11, 12, 17 and 19 of February) where the coverage is sufficient. Then, daily mean sea level pressure maps and wind fields are used to compute the air-sea CO$_2$ flux over the region in a similar way as described in 2.3.1. The salinity maps present major errors near islands, because no correction of the island effect on the SMAP maps (Grodsky et al., 2018) was applied. Therefore, the reconstructed flux will be studied over a region excluding the close vicinity of the islands [59°W-50°W, 5°N-16°N].

## 3 Results

### 3.1 A transition region presenting a strong mesoscale activity

Figure 2a presents the ΔfCO$_2$ measured during the EUREC$^4$A/ATOMIC campaign in January-February 2020. It shows a complex environment that presents similar large-scale features as the climatology, but it reveals numerous smaller scale structures (Figure 1a). Among the latter, two stand out. These are the very low fCO$_2$ in the south-eastern part of the domain, and the high fCO$_2$ around 11°N. Winter is often considered as a period of low variability in the western tropical Atlantic, however, it is still a highly dynamic region. The northwestern tropical Atlantic is commonly divided into two parts, the northern much less variable part (also called "Trade wind region"), and the southern part, also referred as Eddy Boulevard (Stevens et al., 2021). In early 2020, the NBC retroflection was very variable and shed two large anticyclonic rings (Figure 1b,c). They are long-lived 250-km large eddies traveling north-westward toward the Caribbean in the Eddy Boulevard region. The ring detection algorithm TOEddies based on ADT indicates that NBC ring A2 separated from the retroflection in late December, was fairly stationary during the cruise period (February 2020) and located around 58°W-11°N. NBC ring A1 separated from the retroflection in early February 2020 and then stayed around 54°W-10°N for 10 days before translating northwestward toward the Caribbean after the 20$^{th}$ of February. These eddies contribute to the variability of the region in two ways. As they travel, they transport the water trapped in their core during their formation, but they also stir the surrounding waters.

**3.2 Surface water masses identification**

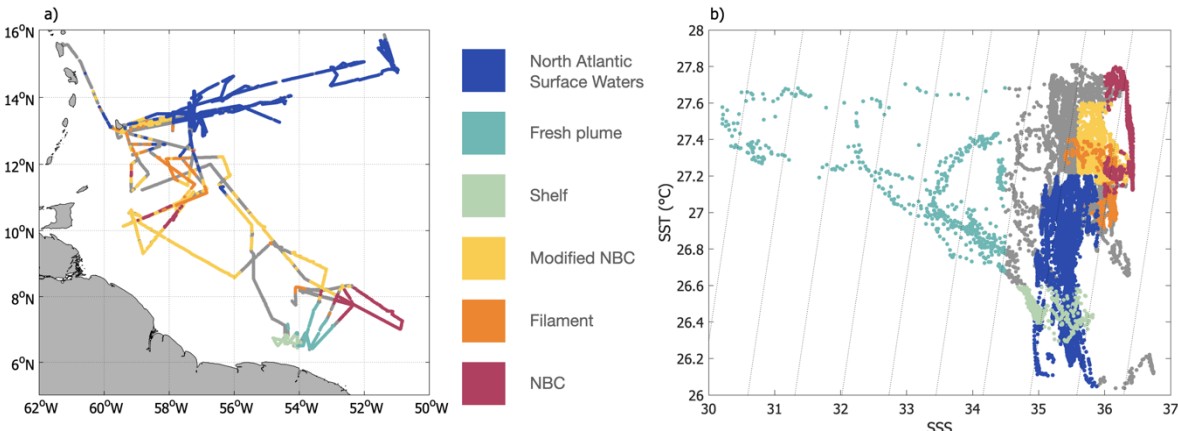

**Figure 4: a) Map representing the RVs Atalante, Merian and Ron Brown ship tracks color-coded with the identified water masses. b) T-S diagram color-coded with the water masses; the gray color corresponds to points that do not fit into the definition of the identified water masses.**


In order to better understand the variability of the region, it is necessary to characterize the main water masses present in winter. From SST and SSS only, we are not able to define water masses with a unique $fCO_2$ signature (Figure 3). By combining SST and SSS with Chl*a* and using information from the dynamical structures of the region, we identified six upper-ocean water masses (Figure 4). For example, the fresh plume water mass corresponds to low SSS, while the NBC water mass is

associated to warm SST, high SSS and low Chl*a* (Figures 3 and 4) in the region of the retroflection and eddy formation. The along track $\Delta fCO_2$ for each ship is presented on Figure 5, color-coded with the identified water mass, highlighting the link between the surface T-S-Chl*a* relation and $\Delta fCO_2$. In this first part, we will present the two surface water masses that are usually identified in the region (e.g. Longhurst et al.,1995, 2010) and their physical properties. We will introduce four new ones in the following parts, as well as their associated dynamic structures.

North of Barbados, the domain is mostly dominated by North Atlantic Subtropical Waters. They have a SSS in the range of 35 to 36 and are relatively cold (SST < 27.2°C). Their SST diminishes over time towards the end of February. These waters are less influenced by coastal dynamics and therefore are not very productive at the surface (Chl*a* levels inferior to 0.14 mg.m$^{-3}$) due to low nutrient levels. They are mainly located north of 13°N, and get progressively colder toward the north-east. The RV Ron Brown stayed in that Trade Wind region for almost a month (Figure 5). The observations collected from this ship

show NASW lower $fCO_2$ with respect to the atmosphere ($\Delta fCO_2 = -40\mu atm$). Similar results are found on one of the RV Merian's transect, with $\Delta fCO_2 < -30\mu atm$.

The surface-intensified NBC is fed by the central branch of the South Equatorial Current (SEC, Schott et al., 1998). As the cold and saline water from the upwelling region is transported westward by the SEC, it warms up (SST > 27°C), but retains its saline characteristic (SSS > 36) as it reaches the NBC retroflection region (Figure 4). This NBC water mass is oligotrophic

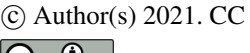


(Figure 3b), and therefore in our area of interest distinguished itself by its low level of surface Chl*a* (Chl*a* < 0.14 mg.m$^{-3}$). These waters are found in the retroflection area, and sampled by both RVs Merian and Atalante at the beginning of February (Figure 5).

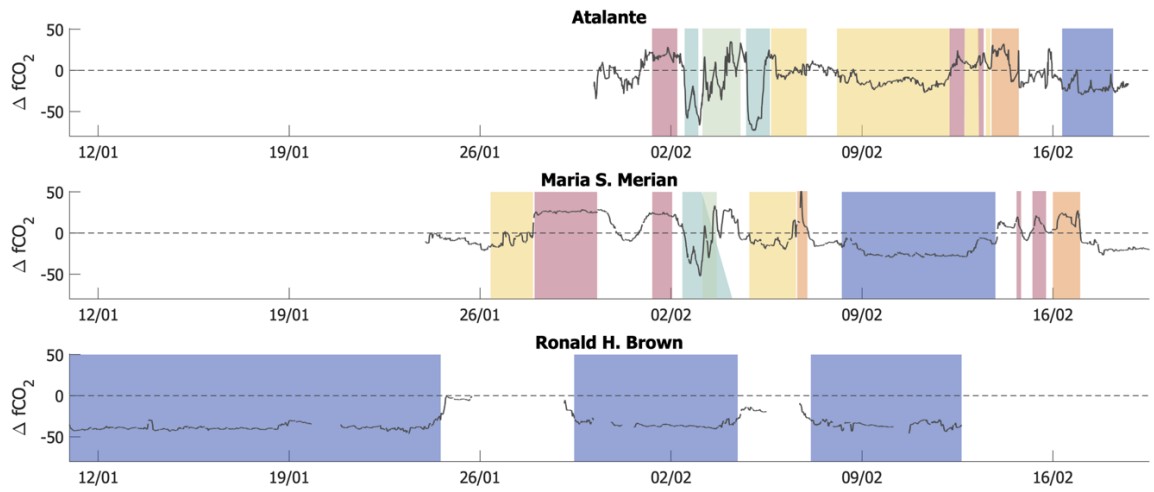

**Figure 5: RVs Atalante (top), Merian (middle) and Ron Brown (bottom) ΔfCO$_2$ time-series. The background color indicates the crossed water mass domains (see definition in legend of Fig 4).**

## 3.3 North Brazil Current rings

The extension of the NBC retroflection varies depending on the state of eddy formation. It moves northwestward up until 9°N as an eddy is forming, and then retracts to the southeastern part of the region. During our period of interest, the retroflection shed anticyclone A1 at the beginning of February. It is difficult to estimate the date of shedding as the area is highly dynamic and detecting the first closed contour of ADT is complicated and may be inaccurate. It is however interesting that the two ships sampled the retroflection when it was expanding to generate A1, and this northwestward expansion is well observed on several

physical and biogeochemical parameters (Figure 6a). RV Merian crossed the retroflection on January, 27$^{th}$ and stayed in the area until February, 2$^{nd}$ (Figure 5). Chl*a* present on the shelf is advected by the strong currents on the periphery of the retroflection and delineates well its south-western side (Figure 6a). The NBC waters stand out on the surface TS diagram, as they are the most saline waters observed in the region (Figure 6b). They are also high in fCO$_2$ which reflects their equatorial origin. Their SST is relatively warm, varying from 27.8 °C at the crossing of the first retroflection front, to 27.2 °C. The region

is rather homogeneous, with an almost constant SSS of 36.3 and ΔfCO2 along the multiple crossings, as observed on the Merian and Atalante transects (Figure 5). On average along those transects, the NBC fCO$_2$ is higher than fCO$_{2atm}$ by 20 µatm.



Anticyclone A1 is further crossed by the RV Atalante on February 6[th], just a few days after its separation from the retroflection (Figure 6c). The surface signal is almost lost, both in SST and in fCO$_2$ (Figure 6cd). It is mainly composed of modified NBC

water, which properties are close to the NBC water (high SSS, high SST, low Chl$a$) but not as pronounced. This water mass covers a larger area, which mainly encompasses the Eddy Boulevard region. It is defined here as SSS > 35.6, 27.16 °C < SST < 27.6 °C and 0.11 mg m$^{-3}$ < Chl$a$ < 0.25 mg m$^{-3}$. While the high Chl$a$ water delimits the retroflection area, it partly covers the eddy A1.

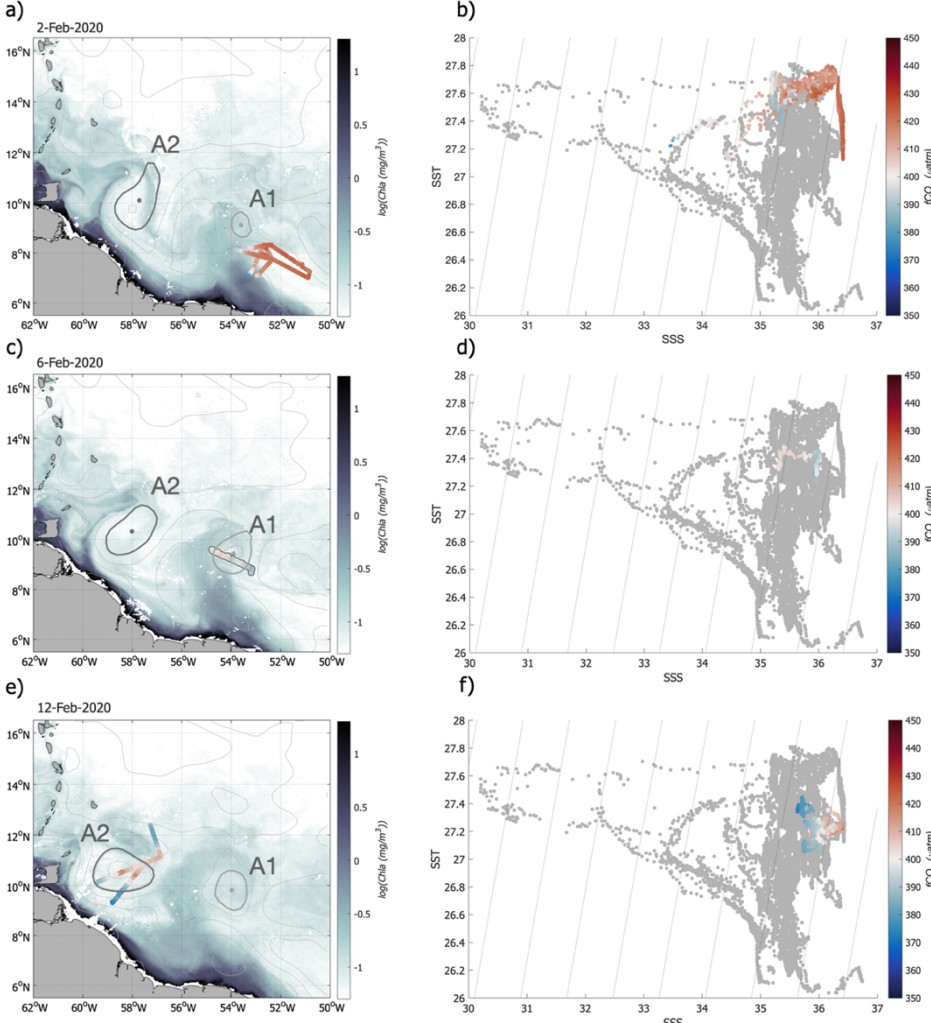

**Figure 6: a) RVs Atalante and Merian ship track in the NBC retroflection (Merian: Jan. 27[th] to Feb. 2[nd], Atalante: Feb 2[nd]), c) in NBC ring A1 (Atalante: Feb 6[th]) and e) in NBC ring A2 (Atalante: Feb 12[th]-13[th], Merian: Feb 13[th]-14[th]) color-coded with fCO$_2$. The background represents the Chl$a$ on Feb 2[nd] (a), Feb 6[th] (c) and Feb 12[th] (e), and the contours of NBC rings A1 and A2 are indicated. b),d),f) Corresponding T-S diagrams color-coded with fCO$_2$.**




NBC ring A2 presents a different situation. Detached from the retroflection in early December 2019 (as defined from altimetry detection), it travelled north-westward while retaining an intense coherent core. Coastal waters identified by their high Chl*a* content were less present and mostly entrained at the north-westward edge of the eddy. After two months, A2 almost reached Trinidad and Tobago and was located around 11°N/58°W when it was sampled by the two ships (Figure 6e). The SST signal

is eroded, and most of the eddy is mostly made of modified NBC water with relatively low $fCO_2$. However high SSS (36.5) and $fCO_2$ (415 µatm) are still visible near the eddy center on the two crossings made by RV Atalante on the 12th and 13th of February. This is confirmed by the two sections of the Merian that crossed the altimetric eddy center and measured SSS 0.5 higher in the 50 km radius around the center and above 36. The NBC water mass is therefore found close to the center of eddy A2, as well as its associated high $fCO_2$.


From the collected observations, it appears that the surface signature of NBC rings is relatively variable and complex. It is well marked in their formation area in the NBC retroflection, where waters brought north by the NBC are warmer, saltier, and higher in $fCO_2$ than the water of the northwestern tropical Atlantic. As the eddies travel northwestward, further away from the retroflection, they may be subject to various processes that modify the surface signal. Unfortunately, the data collected is not

sufficient to shed light on which processes involved in this situation. South of Barbados, away from the retroflection, the modified NBC is therefore the most common water mass. Nevertheless, the NBC water is still sometimes observed months after the separation from the retroflection, in the eddy center.

### 3.4 Freshwater plume

The NBC rings form and evolve in an area highly influenced by the Amazon River plume. Even if February is a period of low Amazon River outflow (Dai & Trenberth, 2002), freshwater events are relatively common. In February 2020, a freshwater plume detached from the Guiana plateau and spread out into the northwestern tropical Atlantic. The off-shelf plume was steered northward by the retroflection and NBC ring A1 up to 12°N and then extended westward toward the Caribbean Sea. Waters carried by the plume strongly contrast with the saline waters of the retroflection. They include water from the Amazon and

present low SSS (SSS < 34.5), low $fCO_2$ ($fCO_2$ < 380 µatm) and high Chl*a* (Chl*a* > 0.25 mg m$^{-3}$). The plume was crossed three times, twice on February 2nd and once on February 5th (Figure 5). Freshwater from the Amazon arrived on the plateau on the 1-2nd of February and was then entrained northwestward by Ekman transport and geostrophic currents (Reverdin et al., 2021). On the 2nd of February RVs Atalante and Merian left the retroflection area to cross the adjacent nascent plume. SSS rapidly decreased, reaching 33 which is associated to a strong decrease of $fCO_2$ (Figure 7). From the 2nd to the 5th of February, the

plume formed and on February 5th the plume is approximately 100 km wide, with lowest salinities around 30. Based on satellite SSS data of the following days, the plume appears to have reached even lower salinities and then spread out over the north-western tropical Atlantic. It first spread northward, steered by A1 and then northwestward, channeled between A1 and A2



reaching all the way up to 12°N and extending over more than 100 000 km² (Reverdin et al., 2021). The plume can be followed
by satellite SSS and Chl*a* maps (Figures 6a, c, e). Indeed, the low SSS is also accompanied by Chl*a* as water from the Amazon
are considered highly productive. The north-western tropical Atlantic is in general nutrient-limited but the nutrients brought
by the Amazon can support the occurrence of a bloom. The plume is also characterized by high silicate (between 4 and 10
µmol kg$^{-1}$ in the plume), while nitrate and phosphate are rapidly consumed. Traces of inorganic phosphorus were observed in
the plume, while nitrates were absent from surface waters (Supplementary Figure S2). Low salinity combined with high
biological productivity led to low fCO$_2$ and a strong carbon drawdown in the plume, as the ΔfCO$_2$ reached -73 µatm on
February 5$^{th}$ (Figure 5).

In an area highly influenced by the NBC waters, through rings or the retroflection, the plume stands out and modify the
biogeochemical dynamics of the region.

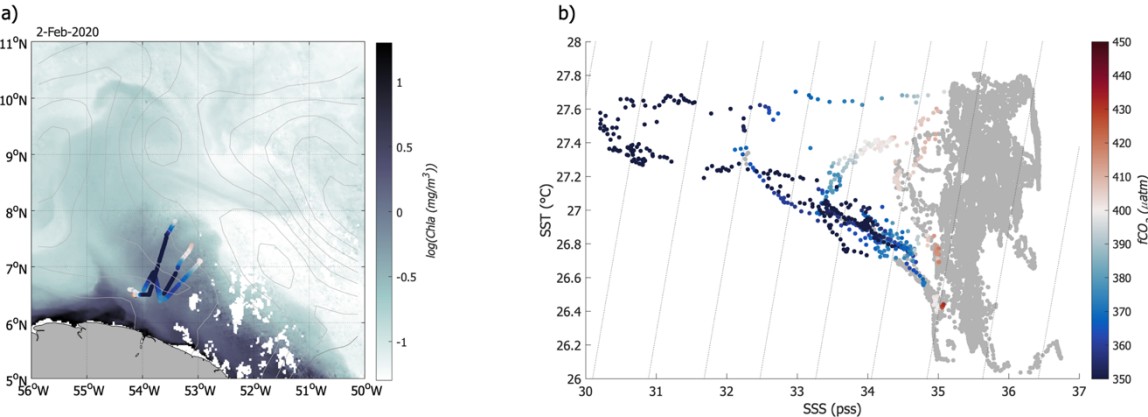

**Figure 7: a) RVs Atalante and Merian ship track in the freshwater plume (Atalante: Feb 2$^{nd}$, Feb 5$^{th}$, Merian: Feb 2$^{nd}$) color-coded
with fCO$_2$. The background represents the Chl*a* on Feb 2$^{nd}$. b) Corresponding T-S diagram color-coded with fCO$_2$.**

### 3.5 Coastal water and filaments

The freshwater plume is not the only water stirred by the NBC rings travelling from the NBC retroflection towards the
Caribbean. The shelf water is very different from the plume water, and was only sampled sparsely on the way in and out of the
plume (Figures 4,5). On the Guiana plateau water is very rich in Chl*a* and detrital material, rather saline (SSS ~ 35.5) and
relatively cold (SST ~ 26.5°C) (Figure 4). Since the water sampled on the edge of the plume was cold due to a local upwelling
event (or vertical mixing event) detailed in the supplementary materials (Figure S3), temperature is not homogenous on the
shelf.

Further north, a filament is stirred on the western side of NBC ring A2 (Figure 8a). It is a small-scale structure, approximately
10 km wide, easily identifiable due to its high Chl*a*. The filament is continuously stirred by A2, and so is already visible on


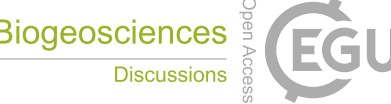

Chl*a* maps of February 2$^{nd}$ (Figure 6). It followed A2 westward's translation and was crossed on February 6$^{th}$ and 17$^{th}$ by RV Merian, and on February 14$^{th}$ by RV Atalante (Figure 5). It has a SSS close to 36, and an SST between 27°C and 27.5°C, thus it is slightly colder and more saline than its surrounding waters (Figure 7b). It stands out by its high Chl*a* content (Chl*a* > 0.25 mg m$^{-3}$), even if this is lower than close to the coast or in the freshwater plume. The strongest signal is observed on the ocean

carbon parameters. In contrast to the freshwater plume, this filament presents very high fCO$_2$ (> 430 µatm), highlighting different origins. It stands out from the ship track time series by also having a larger positive ΔfCO$_2$ (50 µatm). Whereas the freshwater plume observed more southeastward carries water recently arrived on the plateau from the Amazon, the northwestward filament contains shelf waters.

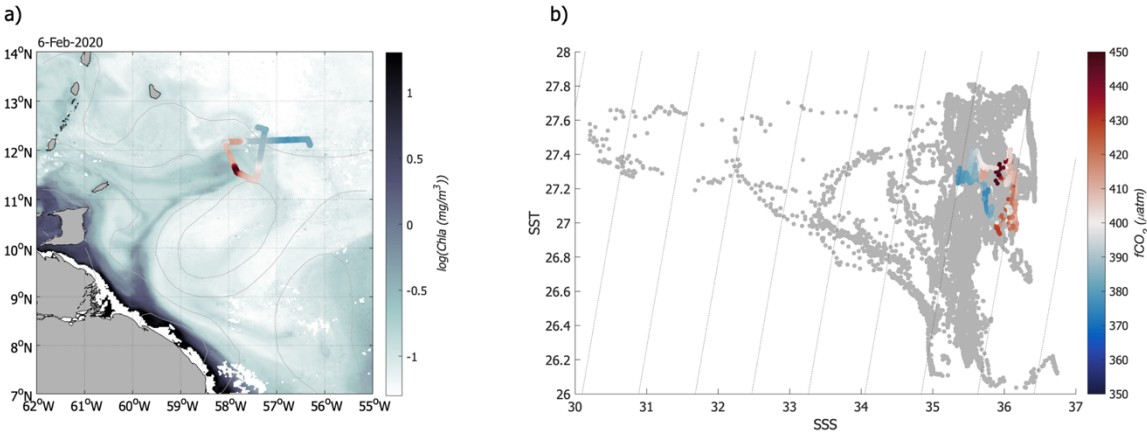

**Figure 8: a) RV Merian ship track in the shelf water filament (Feb 6$^{th}$) color-coded with fCO$_2$. The background represents the Chl*a* on Feb 6$^{th}$. b) Corresponding T-S diagram color-coded with fCO$_2$.**

### 3.6 CO$_2$ air-sea flux

In order to better characterize the impact of each structure on the regional flux, we computed air-sea CO$_2$ maps from satellite

data, at a resolution higher than 60 km (Figure 9), averaged over the period of the cruise (February 2$^{nd}$ to February 19$^{th}$). The along-track flux represented on Figure 9a and the reconstructed regional field (Figure 9b) show the importance of the small-scale dynamical structures, and highlights two strong regimes that are found on the reconstructed map. The air-sea CO$_2$ flux in the northeastern part of the domain, characterized by the NASW, is mainly dominated by temperature effects while further south the presence of NBC rings, and their interactions with shelf waters, create a strong dependence of the CO$_2$ flux on SSS

and on the biological and biogeochemical processes highlighted by Chl*a*.

We evaluate the integrated air-sea CO$_2$ flux over the region. In February, waters are the coldest, and the region is a strong CO$_2$ sink of -1.7 TgC month$^{-1}$ (Figure 10). Three biogeochemical domains mainly contribute to the air-sea CO$_2$ flux, the NASW,



the freshwater plume and the NBC retroflection. The impact on the flux of the small-scale coastal filament is evident along the
ship tracks (Figure 9b). However, its contribution to the total flux is weak as the signal is smoothed when averaging over
February, as the filament moves following the A2 ring northwestward translation. Each of the main three regions is defined
based on its averaged SST, SSS and Chl*a* properties in February and the region-specific flux is determined (Figure 10).

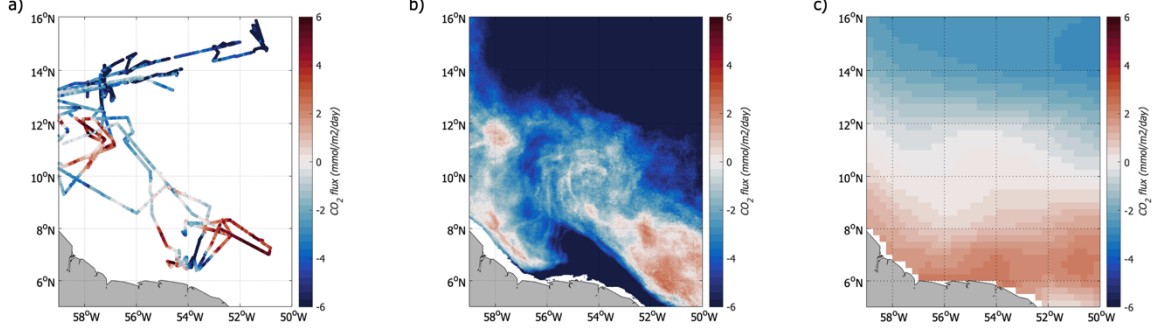

**Figure 9: a) Air-sea CO$_2$ flux measured in Jan-Feb 2020 during the EUREC$^4$A-OA/ATOMIC cruise. b) Air-sea CO$_2$ flux**
**reconstructed over February 2020. c) February climatology of the air-sea CO$_2$ flux over 1998-2015 (Landschützer et al., 2020).**

NASW contributes to about 60% of the total sink due to their relatively cold temperature and to strong winds that enhances
the air-sea exchanges. These waters extend from Barbados northward and eastward, cover more than 1/3 of the domain and
show a weak variability over the month of February.

The NBC retroflection is a source of CO$_2$ to the atmosphere. In February, the strongest signal is observed in the southeastern
part of the domain up to 8°N/53°W. The retroflection nevertheless impacts the region as far as 10°N/54°W as it is spatially
variable, reaching up to 10°N when shedding an eddy. NBC rings present a small positive February air-sea CO$_2$ flux average.
Eddy A1 is almost stationary from its formation date (around February 6$^{th}$) until February 20$^{th}$ and its neutral to slightly positive
CO$_2$ flux is centered around 10°N/54.5°W (Figure 9). Eddy A2 translates rapidly westward at the beginning of February, and
then northward from the 15$^{th}$ of February. Its signal is therefore not as visible as on the ship tracks as it is averaged over one
month. The retroflection is the main region with a positive air-sea CO$_2$ flux, even if the region is too small to have a big global
impact, understanding small-scale features may be significant for the total flux. The NBC rings carry part of the signal, which
is heavily modified as they travel northwestward. As a result, on average for the month of February only the retroflection
maintains a positive flux, while a large part of the domain dominated by modified NBC waters (non-influenced by the plume)
behaves as a small sink.





The freshwater plume with Amazon water is nascent when crossed by the ships (Figure 9b), but is already the strongest signal

of the time-series. As the plume develops, it is entrained by NBC ring A1, then A2 and spreads out into the open ocean, as

observed on SSS and Chl*a* maps. The plume generates a strong $CO_2$ sink that is amplified by strong winds, and reaches up to

12°N (Figure 9). The freshwater plume covers only 10% of the total area, but contributes to almost 20 % of the sink. In winter,

this region is either not characterized in previous studies, or considered as dominated by high $fCO_2$ waters brought by the NBC

on the climatology. We observe here that the direct effect of NBC rings in transporting $CO_2$-rich waters is relatively weak in

winter, and the main signal is associated to the filaments they stir.

The north-western tropical Atlantic therefore behaves as a sink of $CO_2$ in February, driven by the cold north Atlantic

subtropical waters and the Amazon freshwater plume stirred by NBC rings.

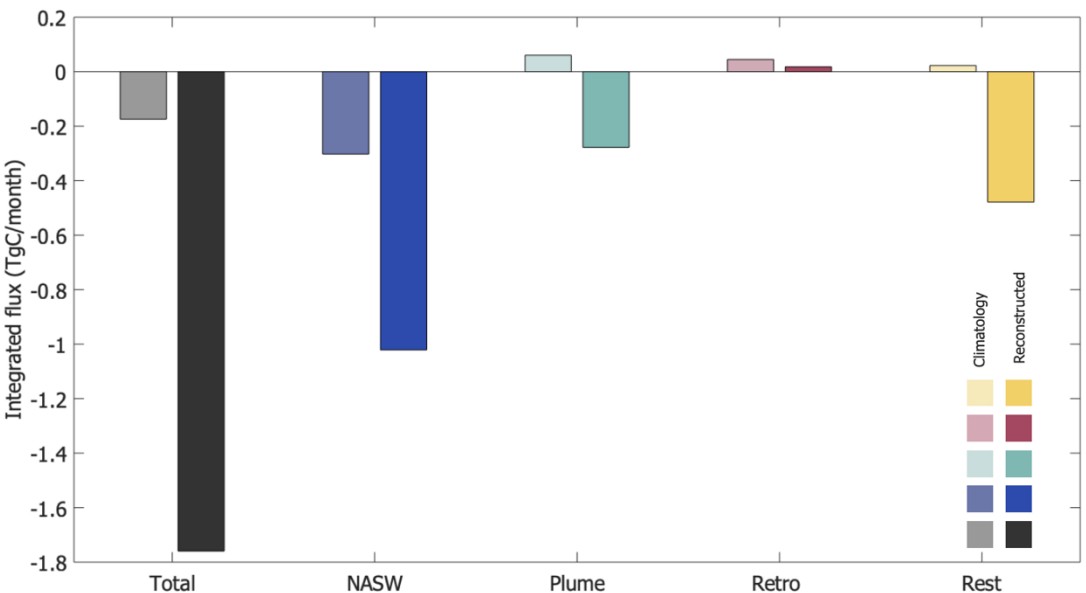

**Figure 10: Integrated flux for the [5°-16°N, 59-50°W] domain, and for 3 water masses. For each bar duet, the one on the left in faded colors represents the integrated flux from Landschützer et al., (2020) February climatology, while the one on the right is computed from the reconstructed flux. Same color code as in Figure 4.**



## 4 Discussion

### 4.1 Biogeochemical provinces

In an effort to understand how biogeochemistry is forced by physical processes in the ocean and atmosphere, we used surface Chl*a* to complement SST and SSS data in defining surface water masses. Partitioning the world ocean into biogeochemical provinces based on physical, geographical and biological criteria has long been proposed to study and better understand biological processes (Fay & McKinley, 2014; Longhurst et al., 1995; Longhurst, 2010). As reviewed by IOCCG (2009), satellite Chl*a* has been widely used to determine biogeochemical provinces, a term coined by Platt et al. (1991) and associated biological primary production but relatively few studies have used such methodologies for mapping air-sea $CO_2$ fluxes. Recently, Landschützer et al. (2013, 2016) have divided the world ocean into biogeochemical provinces for deriving $fCO_2$ and air-sea $CO_2$ fluxes. The way we defined water masses, considering time-varying geographical boundaries, is relatively similar to the one used by Longhurst, and some of the surface water masses compare well with Longhurst (2010) biogeochemical provinces. He identified 3 provinces in the Northwestern tropical Atlantic, The North Atlantic Tropical Gyre province (NATR), the Western Tropical Atlantic province (WTRA) and the Guianas Coastal province. The NATR province is defined as the region north of the North Equatorial Current (12°N-14°N) and east of the lesser Antilles. NATR consistently has the lowest surface Chl*a* of the North Atlantic. The North Atlantic Subtropical Waters of this paper represent the same physical and biogeochemical processes as NATR and we found differences in reference to the WTRA and GUIA provinces. WTRA includes geographically the NBC and modified NBC waters. However, while the retroflection of the NBC is mentioned in the physical characteristics of the WTRA, the eddies are also mentioned in the GUIA province. The GUIA province includes all coastal processes, in this case it can be the freshwater plume, as well as high $CO_2$ coastal waters, and the NBC rings to a certain extent. The latter emphasizes the need to take into account the finer scale in highly dynamic regions.

### 4.2 Integrated $CO_2$ air-sea flux

The northwestern tropical Atlantic present a strong seasonal variability of $CO_2$ air-sea fluxes (Landschützer et al., 2016). In February, waters are the coldest, and the region is a strong $CO_2$ sink of -1.7 TgC month$^{-1}$ (Figure 10). This region, located at tropical latitudes but combining characteristics of subtropical waters and river outflow, is difficult to represent in large scale climatologies. Indeed, the sink for the month of February is smaller by a factor 10 in Landschützer et al. (2020), and is also severely underestimated in Takahashi et al., (2009), but the low spatial resolution of this last product doesn't allow for a good quantitative comparison. This region has been rarely observed, and the interannual variability described in Landschützer et al., (2020)'s climatology is therefore rather uncertain. The compensating effect of different years cannot explain entirely the difference of signal observed in February 2020 with respect to the two climatologies.

Three water masses mainly contribute to the air-sea $CO_2$ flux, the NASW, the fresh plume and the NBC retroflection. The NASW contributes to about 60% of the total sink and are not well captured in climatologies, with noticeable differences of





more than 20 µatm between the measured $\Delta fCO_2$ in 2020 and the one computed from Landschützer et al., (2020) and Takahashi et al., (2009) (the closest grid point is considered for this comparison). The retroflection is the main region with a positive air-sea $CO_2$ flux. Its influence is observed up to 10°N-55°W, but its area is small so its impact on the regional flux is weak. The

positive flux of the retroflection is slightly overestimated in the climatologies, but it could also be due to the difficulty to detect the retroflection at the beginning of February. The main difference is that the NBC waters rich in $CO_2$ are localized in the retroflection area, and are heavily modified when spreading into the Eddy Boulevard.

The freshwater plume is a feature previously not well described for this region in winter and we found a contribution of almost 20 % to the sink. The impact of the Amazon River has been overlooked so far in winter, but it accounts for a large part of the

salinity and biogeochemical variability. Freshwater from the Amazon is not just located on the shelf, but it can spread northward advected by the strong currents variability associated to the NBC rings (Reverdin et al., 2021). These rings are the largest, faster rotating and are the most energetic during boreal winter compared to other seasons (Aroucha et al., 2020). Combined with a seasonal increase in the Amazon's outflow, it induces a large variability in SSS, Chl*a* and $fCO_2$. The occurrence of freshwater export from the shelf to the open ocean has a strong influence on the salinity and therefore on the

mixed layer depth and air-sea heat exchanges (Reverdin et al., 2021). It also strongly impacts the biogeochemistry of the region as the low $fCO_2$ is both due to the low salinity of the plume waters and to the biological activity. The plume stirred into the open ocean by the NBC rings brings nutrients in a region strongly nutrient-limited, and generates a local winter bloom. This in turns plays an important role on the air-sea $CO_2$ flux, and is a crucial feature of the southern part of the northwestern tropical Atlantic.

**4.3 Extension to other years and interannual variability**

Few tropical Atlantic measurements of biogeochemical tracers are available, in particular in the northwestern tropical Atlantic. The EUREC[4]A-OA/ATOMIC campaign provides the first in-situ comprehensive measurements of $fCO_2$ in this region for the boreal winter season. The reconstruction of $fCO_2$ maps likely provides a good understanding of the spatial evolution of $fCO_2$ and air-sea $CO_2$ fluxes, and is fitted for the months of January-February 2020. Although the processes described here are

specific to winter and thus cannot be extended to other seasons, they will be useful to understand the winter variability of other years.

 Only a few cruises cross the region according to the SOCAT database between 2010 and 2019 (period with satellite SSS data) and investigating inter-annual variability is not possible. However, we can test the relation developed for 2020 for other years by using select cruises from the SOCAT database. We thus first use the relationship to reconstruct $fCO_2$ along the ship tracks

(using in-situ SSS and SST and colocalized Chl*a*) and then over the whole region based on satellite products (OSTIA SST, Globcolour Chl*a*, SSS+CCI, detailed in the appendix A). A comparison between the measured and reconstructed $fCO_2$ for the water masses sampled by the SOCAT cruises (NASW, fresh plume, NBC retroflection, modified NBC) is presented in the appendix (Table A1). Good agreement is found between the $fCO_2$ from the SOCAT database and the one reconstructed from the in-situ temperature and salinity, and colocalized Chl*a* for the four water masses (averaged difference of 5.5 µatm). When

comparing reconstructed $fCO_2$ maps with $fCO_2$ on ship tracks (Figure A1), the agreement between $fCO_2$ in various water masses is very clear, even though the spatial structures are sometimes a bit misplaced. This is attributable to the slightly coarser resolution of satellite products not designed specifically for each campaign, to the high spatio-temporal variability of $fCO_2$ and to missing Chl$a$ and SST observations in cloudy areas. February 2020 was mainly cloud free, so that we were able to use high resolution daily SST and Chl$a$. The SSS product used in 2020 is also a daily product. However, for the other years, the satellite
CCI Chl$a$ (if clouds) and SSS products have a weekly temporal resolution, which smear the fast-moving structures. The gradients between water masses are therefore not always well represented, but we find a good agreement between the $fCO_2$ of each structure, which is encouraging for future studies on interannual variability in winter.

By identifying the main processes responsible for the variability of the air-sea $CO_2$ flux in 2020, we can better understand the
interannual variability of the region. Indeed, each of the main water mass has its own interannual variability that shapes the $CO_2$ variability. The northern part of the domain is dominated by the variation of temperature, and therefore its interannual variability is linked to the one of SST. From 32 years monthly mean SST data, the SST standard deviation in the area is relatively weak, and doesn't exceed 0.5°C. The northern sink of $CO_2$ is therefore rather similar from year to year, coherent with the low standard deviation of the air-sea $CO_2$ flux computed from Landschützer et al., (2020).
As previously stated, most of the variability occurs south of Barbados. The freshwater plume sampled during EUREC[4]A-OA is a common feature in February. During the 2010-2019 period, events of freshwater reaching the open ocean were observed each year, and freshwater plumes similar to the one described in this paper were observed during 7 out of 10 years from satellite salinity data (Reverdin et al., 2021). Two of the main mechanisms driving the occurrence of the plume are the winds near the Amazon estuary that can induce along shelf transport to the Guyana plateau and the presence of NBC rings. This region is
commonly crossed by several NBC rings during winter (Jochumsen et al., 2010; Johns et al., 2003; Mélice & Arnault, 2017) but it also is subject to a strong year to year variability that has linkages with the variability of the Amazon River outflow (Aroucha et al., 2020). Therefore, identifying and understanding the processes happening in 2020 should contribute to better assess the interannual variability of $fCO_2$ as well as air-sea $CO_2$ fluxes in the northwestern tropical Atlantic during winter. Using a combination of SSS, SST and Chl$a$ brings information on the biogeochemistry of the area in winter and represent well
the mesoscale structure.

## 5. Conclusion

The EUREC[4]A-OA/ATOMIC campaign provides for the first time synoptic measurements related to the air-sea fluxes of $CO_2$ in the northwestern tropical Atlantic in winter. Six main surface water masses are identified, one of them north of Barbados (North Atlantic Subtropical Water), and the other five (the NBC retroflection, modified NBC waters, the freshwater plume,
the shelf water and the shelf filament) south of Barbados. The investigation highlights the two different regimes of the region. In the northern part, the variability of the $CO_2$ flux is low and the area is covered by relatively cold, saline and low-chlorophyll





NASW. The southern part is highly variable, due to the presence of large mesoscale anticyclonic eddies. In January and February 2020, two NBC rings influence the physical and biogeochemical properties of the region. The NBC retroflection is characterized by waters with equatorial origins that are relatively warm, saline and high in $fCO_2$. As the rings separate from

the retroflection, they interact with the surrounding waters, and the initial signal in $fCO_2$ is dampened. The main impact of the rings is therefore not necessarily on the surface water they transport in their core, but rather on the filament they stir off the coast. A fresh plume from the Amazon River is transported by the coastal current up to the French Guiana shelf in the beginning of February. The NBC rings entrain the plume of freshwater up to 12°N. This plume is fresh, rich in Chl*a* and low in $fCO_2$ and strongly contrasts with the surrounding waters and spreads over ~100 000 km². On the shelf not influenced by the plume water

is relatively saline, high in $fCO_2$ and Chl*a*, probably due to high concentration of detrital material. As ring A2 propagates westward, it continuously stirs a thin (10 km wide) filament of high $fCO_2$ shelf water up to 12°N.

Based on the ship observations we identify distinct regime in $fCO_2$ linked to certain combinations of SST, SSS, and Chl*a* properties. We use this information to construct high-resolution maps of $fCO_2$ and air-sea $CO_2$ flux using satellite maps of

SSS, SST and Chl*a*. On average over the month of February, the region acts as a strong sink of $CO_2$ (-1.7 TgC/month), the sink being five times smaller in air-sea $CO_2$ flux climatologies. The NASW contributes for most of the flux (60%) due to low temperature associated to winter cooling and strong winds. South of Barbados, the region acts also as a sink of $CO_2$. The influence of equatorial water is localized to the retroflection region that acts as a small source of $CO_2$. The main feature in this part of the domain is the fresh plume that contributes to almost 20% of the total sink.

The processes described here highlight the high variability of air-sea $CO_2$ fluxes in winter, that are quite different from the ones in summer. These features are relatively common in winter and can be used to better understand the interannual variability of air-sea $CO_2$ fluxes. The northern part of the domain is driven by the variability in SST, while the southern one is a combination of the interannual variability of temperature, salinity and chlorophyll. It is therefore linked to the year-to-year variability of the NBC rings and the Amazon outflow.

This study is limited by the paucity of data in the region and for this time period. More $fCO_2$ data closer to the coast would help to better quantify the influence of shelf water on the flux. The signature of the NBC rings has been described for only two rings that had different signatures. In order to reach more robust conclusions on the transport of surface NBC water by the rings, more eddies should be observed. The variability of $fCO_2$ occurs at large and small scale. Salinity is one of the most valuable predictors of $fCO_2$ south of 10°N, but the satellite salinity resolution is much lower than the one of temperature and

chlorophyll. To have a more accurate prediction of the $fCO_2$, a high-resolution SSS product would also be very useful.



**Appendix A**

Due their long time series, the following SST, Chl*a* and SSS products are used to reconstruct fCO$_2$ maps in winter in the northwestern tropical Atlantic for other years than 2020. Results are shown on Figure A1. They are different than the satellite products used in the main study, that were only available on a short period.

The Operational Sea Surface Temperature and Sea Ice Analysis (OSTIA) SST product, distributed by the CMEMS is used here. Daily maps of SST are produced at a resolution of 1/20°, available from 1981 to present. OSTIA SST uses most SST data available for a day, from both infrared and microwave inferred SST.

Surface Chl*a* from GlobColour dataset derived from ocean color at a 1/24° resolution is used. It is a merged product from multiple satellite missions' observations (SeaWiFS, MERIS, MODIS, VIIRS NPP, OLCI-A, VIIRS JPSS-1 and OLCI-B).

GlobColour data is developed, validated, and distributed by ACRI-st.

We also use SMOS and SMAP combined weekly SSS generated by the Climate Change Initiative Sea Surface Salinity (CCI + SSS) project (https://doi.org/10.5285/4ce685bff631459f- b2a30faa699f3fc5). It provides weekly level-3 SSS data from 2010 to 2019 at a spatial resolution of 50 km, a sampled on a 25 km x 25 km grid, and 1 day, by combining data from the SMOS, Aquarius, and SMAP missions.


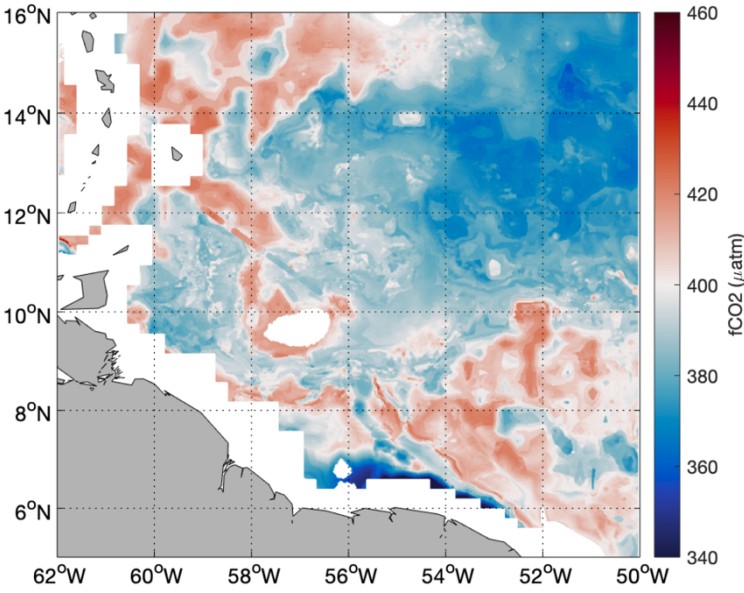

**Figure A1. fCO$_2$ reconstructed from OSTIA SST, CCI+SSS and Globcolour Chl*a* for the 23/12/2015 superimposed with the fCO$_2$ from cruise 642B20151209.**





| | **Fresh plume** | **NBC waters** | **Modified NBC** | **NASW** |
|---|---|---|---|---|
| **SOCAT fCO₂** | 316.2 | 413.4 | 385.8 | 349 |
| **fCO₂ reconstructed from SOCAT SST & SSS** | 310.7 | 410.7 | 392.9 | 358.7 |
| **Transect date** | 2016/01/05 | 2016/01/08 | 2015/12/23 | 2013/02/10 |
| **Ship Name** | Colibri (France) | Colibri (France) | MSC Marianna (Panama) | Benguela Stream (Netherlands) |
| **Expocode** | 35MJ20151229 | 35MJ20160107 | 642B20151209 | 33RO20130108 |

**Table A1.** Comparison for the 4 main water masses between the $fCO_2$ from SOCAT transect and the $fCO_2$ reconstructed from in-situ SSS and SST and colocalized Chl*a*.

## Code Availability


Code used in this study can be made available upon reasonable request to the corresponding author.

## Data Availability

We benefited from numerous data sets made freely available and listed here: the ADT produced by Ssalto/Duacs distributed by CMEMS (https://resources.marine.copernicus.eu), the Chl*a* and SST maps produced by CLS

(**https://datastore.cls.fr/catalogues/chlorophyll-high-resolution-daily** and **https://datastore.cls.fr/catalogues/sea-surface-temperature-infra-red-high-resolution-daily**), the SMOS L2Q field produced by CATDS (CATDS, 2019) (https://10.12770/12dba510-cd71-4d4f-9fc1-9cc027d128b0), the SMAP maps produced by Remote Sensing System (RSS v4 40 km), the CCI+SSS maps produced in the frame of ESA CCI+SSS project (**https://10.5285/4ce685bff631459fb2a-30faa699f3fc5**), the OSTIA SST and Copernicus -GlobColour Chl*a* distributed by the CMEMS

(SST_GLO_SST_L4_REP_OBSERVATIONS_010_011 and OCEANCOLOUR_GLO_CHL_L4_REP_OBSERVATIONS_009_082).

The RV Atalante $fCO_2$ is available on the SEANOE website: doi/10.17882/83578. The RV Ron Brown and RV Merian $fCO_2$ data can be found on the SOCAT database (expocodes 33RO20200106 and 06M220200117 respectively). The Surface Ocean

CO$_2$ Atlas (SOCAT) is an international effort, endorsed by the International Ocean Carbon Coordination Project (IOCCP), the
Surface Ocean Lower Atmosphere Study (SOLAS) and the Integrated Marine Biosphere Research program, to deliver a
uniformly quality-controlled surface ocean CO$_2$ database. The many researchers and funding agencies responsible for the
collection of data and quality control are thanked for their contributions to SOCAT.

## Author contribution

LO, JB, GR and NL conceptualized the project. LO carried out the measurements and data analysis. LO, JB, GR, NL and PL
contributed to result interpretation. PL, MR and RW provided the crucial datasets. LO, MR, SS and JK conducted field work.
LO wrote the manuscript with input from all co-authors.

## Competing interests

Some authors are members of the editorial board of Biogeosciences. The peer-review process was guided by an independent
editor, and the authors have also no other competing interests to declare.

## Acknowledgements

This research has been supported by the European Research Council (ERC) advanced grant EUREC$^4$A (grant agreement no.
694768) under the European Union's Horizon 2020 research and innovation program (H2020), with additional support from
CNES (the French National Centre for Space Studies) through the TOSCA SMOS-Ocean, TOEddies, and EUREC$^4$A-OA
proposals, the French national program LEFE INSU, by IFREMER, the French research fleet, the French research
infrastructures AERIS and ODATIS, IPSL, the Chaire Chanel program of the Geosciences Department at ENS and the
EUREC$^4$A-OA JPI Ocean & Climate program. LO was supported by a scholarship from ENS and Sorbonne Université. We
thank Jonathan Fin at the Service National d'Analyse des paramètres Océaniques du CO$_2$ (SNAPO-CO$_2$) at LOCEAN for the
analysis of DIC and TA samples, and François Baurand at the US IMAGO for the nutrient analysis. We thank Matthieu Labaste
and Christophe Noisel for the help with the CO2 measurements onboard RV Atalante. Kevin Sullivan performed the data
reduction and quality control of data on the Ronald H Brown. We also warmly thank the captain and crew of RVs Atalante,
Maria S. Merian and Ronald H. Brown. The measurements on the Ronald H. Brown were supported by the Global Ocean
Monitoring and Observation (GOMO) program (fund Ref. 100007298).

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
