# Peer review of "Wintertime process study of the North Brazil Current rings reveals the region as a larger sink for CO2 than expected"

_Biogeosciences, 2021_

## Author Comment (AC1)

**Response to RC1: Referee #3**

Dear editor and referee. Thank you for the very positive feedback and constructive comments on the manuscript. We have modified the manuscript in order to address your comments. Hopefully, the manuscript will appear better organised in its revised form. Our reply to your comments follows in *italic font* below.

**Overview**

The authors conduct a case study on the air–sea flux in the equatorial Atlantic. They collected pCO2 measurements from several cruises in the under sampled December-February period. They characterise the different water masses they sampled. They develop a relationship to calculate regional pCO2 fields which they use to calculate the regional flux. They identify the flux as being an order of magnitude greater than the Landschützer estimate. They identify the key processes contributing to this flux. This was a really nice paper to read and I found the results interesting. I would suggest publishing this paper following minor revisions.

**Major comments**

**Throughout** – At the moment many of your figures are too small. All axis ticks and axis labels need to be much bigger. There are a lot of different colour schemes throughout the figures, e.g. Figure 1 has three different colour schemes. Can they be standardised? Otherwise it is less clear when the colours refer to the regions in figures 4 and 5.

*We agree, and all the figures' labels and ticks have been made bigger. Figure 1 has been modified and separated into two figures. We included a panel with the SST on Feb $6^{th}$ and chose the same colour scheme for SST and SSS in an effort to standardise. We still retained different colour schemes for fCO2 and Chla in order to differentiate these variables.*

**Title** – It would also be nice for the title to reflect the main finding of the paper. At the moment you don't say what the main result was. For example something along the lines of - pCO2 measurements made in the North Brazil Current rings in February 2020 reveals the region as a larger sink for CO2 then predicted by pCO2 climatologies". I am not suggesting using this title but hopefully it points you in the right direction.

*We propose:*

*Wintertime process study of the North Brazil Current rings reveals the region as a larger sink for $CO_2$ than expected.*

**Abstract** – The abstract needs a bit of work. You have identified the key processes but it isn't clear when and where they are important. This information is in the text but the abstract needs to stand up on its own. You also need 1 sentence at the beginning saying the wider context of the problem, 1 sentence explaining what these eddies are/ how they form and 1 sentence saying how they might impact the flux. Or something along those lines. I would advise using the nature abstract template as a guide on how to improve this section. https://unl.libguides.com/c.php?g=51569&p=2633458

*Thank you for the reference, we followed the nature abstract guidelines to propose the new following abstract:*

*The key processes driving the air-sea $CO_2$ fluxes in the western tropical Atlantic (WTA) in winter are poorly known. It is a highly dynamic region, with expected large role of ocean physics on the variability of $CO_2$ air-sea flux. In early 2020, this region was the site of a large in situ survey which was put into a wider context through satellite measurements. In this season, the North Brazil Current (NBC) flows northward along the coast of south America, retroflects close to 8°N and pinches off the world's largest eddies, the NBC rings. The rings are formed to the north of the Amazon River mouth, which freshwater export is still significant in winter, despite being a period of relatively low runoff. We show that in February 2020, the region [50°W-59°W – 5°N-16°N] is a $CO_2$ sink from the atmosphere to the ocean (-1.7 TgC.month$^{-1}$), a factor of 10 greater than previously estimated. The spatial distribution of $CO_2$ fugacity is strongly influenced by eddy stirring south of 12°N. During the campaign, a nutrient rich freshwater plume from the Amazon River is entrained by a ring from the shelf up to 12°N leading to high phytoplankton concentration and to a significant carbon drawdown (~20 % of the total sink). Trapping equatorial waters, the NBC rings themselves are a small source of $CO_2$. The less variable North Atlantic subtropical water extends from 12°N northward. They represent ~60 % of the total sink due to their lower temperature associated with winter cooling and strong winds. Our results, in identifying the key processes influencing the air-sea CO2 flux in the WTA, highlight the role of eddy interactions with the Amazon River plume. It sheds light on how the previous lack of data impeded a correct assessment of the flux, and on the necessity of taking into account features at meso and small scale.*

**Introduction** – A well written introduction. The thing I feel is really missing here is a full size schematic of the region. You need to label all the currents (with arrows) and locations you mention in the introduction. Unless the reader is extremely familiar with the region they will not be able to visualise anything. This becomes important later when you start discussing the cruise track. On a read through it would be great for you to at least label the following North Equatorial Counter Current, NBC retroflection region, the Caribbean, Lesser Antilles, Amazon River plume, Trade wind region, Eddy Boulevard. Currently figure 1 is not that helpful. You show these fields but without a schematic of the circulation/currents it isn't particularly useful. Please add a locations/ currents map as a large subplot to figure 1 at least.

*Thank you for the suggestion. We now dedicate one Figure to a schematic figure (see below), where we describe the circulation and add the name of most places discussed in the paper on top of a snapshot of SSS and of the Landschützer20 climatology.*

[Figure]

**Figure 1: Schematic of the main ocean currents in the western tropical Atlantic superimposed over the SSS field of Feb. 7th 2017 (a) and over the February ΔfCO₂ climatology from Landschüter et al., 2020 (b).**

**Methods** – You need to provide a date for the data, at the moment you state January and February. Either as another subplot by day of year or state them in the text and figure cation. As it stands we don't know how far apart the measurements were made from each other, they could be 1 week or 8 weeks. Figure 2a - c are very much results, so move them there. Perhaps have a cruise track by DOY here instead. Then move Figs 2a-2c to the results. Section 2.2 is nicely written. I feel there is some missing detail here on justifying the choice of datasets though, you need to rationalise why you have used SMAP over ESA CCI SSS for example. In section 2.3.2 you verge into results and begin to discuss them. Be careful of doing this. See lines 214 to 226. I realise you use fig 3 to get the relationship you use to generate your maps but consider moving it to the results.

*The cruise dates for all ships have been included in the text, and we added as suggested a panel with the cruise track coloured with the DOY. We detailed a bit more our choice of the salinity product in the manuscript. The SMOS/SMAP blend from Reverdin et al., 2021 used in this study has been developed especially for this cruise at a shorter temporal resolution than the ESA CCI SSS product. It provides the salinity at a temporal resolution of 1 day, combining 6 am and 6 pm measurements. The ESA CCI SSS has a temporal resolution of 7 days, therefore is less adapted to study the fast-varying Amazon plume in our study.*

*A part of 2.3.2 has been moved to results, as well as Figure 3 that has been combined with Figure 4, thank you for the suggestion.*

**Results** – If you talk about Figure 1 here bring it down to results. Figure 4 and 5 are really great. It would be nice if you had a table describing the criteria for each of the 6 water masses. This will make it easier to quickly reference backwards and forwards.

Please number and or define the water masses as you introduce them. It isn't clear whether lines 290 to 203 describe some of these water masses. Section 3.3,3.4,3.5,3.6 are nicely done. Figure 10 is also really nice, maybe you can discuss this further.

*Figure 1 has been moved to results. Thank you for the nice comments on the figures. A table with the water mass criteria is a great idea and is now added.*

| | NASW | NBC | Modified NBC | Freshplume | Shelf | Filament |
|---|---|---|---|---|---|---|
| Temperature (°C) | <27.2 | > 27 | 27.16 < SST < 27.6 | | < 26.6 | < 27.4 |
| Salinity | 35 < SSS < 36 | > 36 | > 35.6 | < 34.5 | | 35.8 < SSS < 36.3 |
| Chlorophyll-a (mg.m$^{-3}$) | < 0.14 | < 0.14 | 0.11 < Chl$a$ < 0.25 | > 0.25 | > 0.25 | > 0.25 |

**Table 1.** Thresholds in SSS, SST and Chl$a$ used to define the 6 water masses identified.

**Discussion** – Section 4.1 seems out of place. Combine with section 3.2 at the start of the results where you define the water masses. In lines 477 and 478 you say that the inter annual signal can't explain the entirety of the differences, this is a really important point but you don't back this up with hard evidence. Did you try to do an extrapolation for any other years? The appendix figure and table are proof of the method working and I feel justify their place in the main manuscript. The appendix figure is not clear, there doesn't appear to be anything superimposed on it? What is missing is a 10 year timeseries of the winter pCO2 fields using your relationship, it would be nice to visualise the interannual flux variability as a bar chart (maybe split by your 6 regions).

*As suggested, we combined part of section 4.1 with section 3.2. Regarding the interannual signal, we decided to add a Figure presenting the fCO2 reconstructed at chosen dates over the last 10 years (see below). They are chosen to highlight the variability of the fCO2 in the region linked to the interactions between the NBC ring and the Amazon plume. The figure illustrates the different fresh plumes observed for different years and the impact on the fCO$_2$. It also shows the smaller variability of the northern part. The in-depth study suggested here is a very good idea, and will be the object of future research. The appendix figure has the fCO$_2$ measured by the ship superimposed over the map of reconstructed fCO2. They matched quite well and so were hard to distinguished. We added a black contour to delineate the ship track.*

[Figure]

**Figure 2: Snapshot of reconstructed fCO2 for all occurrences of fresh plumes extending at least to 10°N and east of 56°W in January-March 2010-2019 (2010, 2011 and 2013 do not present this type of event).**

**Minor comments**

- Line 1 - The title should read as "The" Impact of North Brazil Current rings on air-sea CO2 flux variability in winter 2020.

  *The title has been modified.*

- Line 13 – this should just be pass

  *This line has been changed*

- Line 19 – factor of 10 might be changed. You could also say underestimated by 1 order of magnitude. You also need to be clear this is WRT to L20 climatology.

  *Done*

- Line 22 – is this 20% of the -1.7Tg month? May say the size as well as the percentage?

  *It is 20% of the -1.7 Tg month$^{-1}$*

- Line 23 - The spatial distribution of

*Done*

- Line 23 – 'On the other hand' is not really the right term here. You mean 'in contrast'. Where is the detrital material acting as a CO2 source?

  *We removed this part from the abstract. The detrital material from the shelf stirred into the open ocean acts as a $CO_2$ source.*

- Line 24 – It is not clear why 12 degrees north is an important threshold from the abstract.

  *Barbados island at ~12°N approximately delimits the trade wind influenced region from the NBC influenced one.*

- Line 30 –"retroflects" is a term I've never heard before. Be aware that some readers even with a strong oceanography background may not be familiar with this. I strongly recommend explaining it here or using another description like " doubles back on its self".

  *We added a definition of the word in the introduction.*

- Line 35 – Is the NBC retroflection region where the rings pinch off or where they travel across?
  *The retroflection pinches the rings: this is mentioned in more details on line 60.*

- Line 37 – For something extensively studied is there not more recent literature? A 2002 reference predates satellite salinity
  *It is a valuable comment: we cite papers on the NBC rings up until 2020 (Aroucha et al., 2020, line 73). Line 37 emphasizes the important work done during the NBC ring experiment period, described in the paper of Wilson et al, 2002.*

- Line 52 – please edit this, the way it is written makes it sound like the ocean is causing ocean acidification.

  *True, thank you, we modified.*

- Line 54 – remove "continuously". It is technically not correct as there is a seasonal signal each year that reduces atmosphere global pCO2.

  *Indeed, thank you for noticing.*

- Line 71 – edit needed. "The minimum plume extend occurs ". Please be careful when referring to seasons near the equator, it may be better to do month X–month Y.

  *Agreed, replaced by month X to month Y.*

- Line 109 – why is this separate paragraph. You can merge it with the paragraph before.

  *Done*

- Line 127 – Should this not reference Takahashi 1993?

  *This references the online pCO2 dataset used in this study (Olivier et al., 2020). We modified a bit the sentence to avoid confusion, and added the reference to Takahashi 1993.*

- Line 128- Can you state the style so the reader doesn't have to look it up e.g. showerhead, membrane, bubble?

  *Done*

- Line 133/134 – You need to describe the methods here. At least one reference to another paper is required.

  *We added this information to the manuscript: DIC and TA were measured at the SNAPOCO2 by potentiometric titration using a closed cell, following the method of Edmond (1970). Nutrients were conserved by heat pasteurization and analysed by colourimetry at IRD LAMA service in Brest.*

- Line 141 – 142 – Can you provide a reference here please
  *Reference added: Tennekes 1973*

- Line 145 - replace "inferior" with "less than" . Also was this comparison at the same time? Hours apart days?
  *Replaced. It was at the same time, we added this information to the text, and it is now visible on the new Figure 2 with the DOY period.*

- Line 154 – How did you check this? Please add the details here or the supplement

  *Thank you. We checked it by removing a bias of 6muatm on the Merian data and reconstructing again the flux over our region. We then analysed the differences between the maps with and without the bias. We added the information in the text.*

- Line 170 – convention to not use PSU or pss. Please check throughout for this

  *Checked and modified, we still mention that practical salinity is used in this study.*

- Line 243 – Would be really nice for you to give the relationship here. Other researchers may want to use it.

*The method does not produce a direct relationship but rather a mathematical object called "interpolant" that can be used as to compute the fCO2 for a combination of T,S,Chla.*

- Line 258 – it is hard to visualise this grid without drawing it on a map

   *Thank you for the interesting suggestion, this grid is interesting to observe in 3D indeed.*

- Line 264- Comparisons with the Landschützer product would be best saved for the discussion.

   *Thank you for the comment, most of the comparison with the Landschützer product is indeed saved for the discussion.*

- Lines 265 -268 and 274-275– This is introductory material. Why is it in results?

   *This part describes the dynamics of the region, it is indeed a bit introductory and we moved some of it in the introduction.*

- Line 302 – Can you change your dates to mmm-dd. Some American readers are confused by dd/mm.

   *Done*

- Line 507 – Are there really only 4 cruises over this time period?

   *We chose one cruise per water mass we described. For some water masses (such as the shelf one), no cruise was available*

---

## Author Comment (AC2)

**Response to RC2: Peter Land**

Dear editor and reviewer. Thank you for the positive feedback and constructive comments on the manuscript. We are glad that you enjoyed the work presented here. We have modified the manuscript in order to address your comments and the ones of Referee #3. Hopefully, the manuscript will appear better organised in its revised form. Our reply to your comments follows in *italic font* below.

This is a thorough and well-presented analysis of a multi-ship campaign measuring waters north of Brazil containing two large eddies (rings), combined with satellite data (including a novel SSS product) to estimate the February air-sea CO2 flux in the region and the contributions due to the rings and other water masses. It describes a method for estimating flux that could be extended to other years, and validates this using previous cruise data with good agreement. I consider the work novel and important and the argument convincing, and I recommend publication. I have two main issues plus detailed comments.

SST is a fundamental component of your method, albeit with relatively low variability across the region, and is often referred to in the text, but we only ever see SST in TS plots and cruise tracks. I would like at least a February 2020 SST composite so readers can see how features described in the text manifest spatially in SST. You could go further and expand other figures to include maps of SST alongside SSS and chla. Speaking of maps, I second Referee #3's request for a geographically-labelled map, including Bermuda, Trinidad & Tobago and any other geographical features mentioned in the text.

*We agree, it is a good suggestion. In order to address this issue, we added a geographically labelled map of a snapshot of SSS and of the CO2 climatology (see below). Then, on a separate figure we added the SST map for the 6th of Feb 2020, together with the Chla and SSS maps for the same day.*

[Figure]

**Figure 1: Schematic of the main ocean currents in the western tropical Atlantic superimposed over the SSS field of Feb. 7th 2017 (a) and over the February ΔfCO₂ climatology from Landschützer et al., 2020 (b).**

In Section 4.3 and Appendix A, you appear to get to a point where you can address a crucial question, which is whether 2020 was a flux outlier, but you don't present any results! I couldn't understand why that was, the paper would be much better with this information. Are you saving it for another paper? I wouldn't blame you, but then Appendix A seems largely redundant as well as unfair on your expectant readers! You successfully validate in other years with cruise data but you don't go on to process all years since 2010 and present summary statistics on at least the headline regional flux figure.

*Thank you for this nice comment. We indeed thought the results would go beyond the scope of this paper and might be the subject of some ulterior study. We decided to added one figure to illustrate the strong interannual variability of the fCO2 in the region linked to the interactions NBC rings – Amazon plume as observed in 2020 (see below). It also highlights changes in the north-eastern part of the domain, but in order to interpret this variability we would need to place it in a longer time scale context. We can nevertheless see the impact of a strong negative SST anomaly in March 2014.*

[Figure]

**Figure 2: Snapshot of reconstructed fCO2 for all occurrences of fresh plumes extending at least to 10°N and east of 56°W in January-March 2010-2019 (2010, 2011 and 2013 do not present this type of event).**

l18-9 This could use more neutral language, eg ' a factor of 10 greater than previously estimated'

*We agree and modified the sentence.*

l22 causes

*Done*

1l39 Different families of rings exist... (a little more explanation of the families would be nice here for ignorant folks like me, e.g. are they all anticyclonic? I would have thought so, but the next phrase kind of suggests a secret world of cyclonic eddies undetectable by altimetry - maybe I'm letting my imagination get the better of me!)

*It is a very interesting subject indeed. The eddies called NBC rings are all anticyclonic. Some cyclonic eddies are also present in the region, but they are not long lived. The different families refer to their depth, some rings are deep and not surface intensified, while the rings with a surface signal are most of the time shallow. The sentence has been changed to avoid confusion and to better explain the subject.*

l58 $CO_2$-undersaturated

*Done*

l72-3 The climatology of difference...

*Done*

l78 ...due to onshore winds as it travels... (as it stands it sounds like winds ambiguously perpendicular to the coast are travelling NW!)

*Thank you, done*

l83 Later you only refer to the western one as a filament and to the eastern one as a plume - be consistent please

*Indeed, we chose to refer them differently to avoid the confusion between the two in the manuscript. Moreover, the scales of the two filaments are different, the eastern one is almost a mesoscale feature (100km wide) so is referred as a plume.*

l124 Temperature and salinity...

*Done*

l133-4 At what depth? If similar to the intake, one thing these might be used for is to

shed further light on the fCO2 comparison (or muddy the waters!) by calculating fCO2.

*Some of them were surface, but none was taken in the shelf waters. We compared measured fCO2 on RV Atalante with DIC and Alkalinity samples, which confirms that the Atalante data is correct. However, it doesn't bring additional information as it was taken along track.*

l169 Was the extra data only used to fill gaps, or were all data from the three passes averaged? If the latter, for consistency I'd be tempted to treat all days the same, and either use 6AM the next day all the time or not at all. Could the two missing days be recovered by using 6PM from the previous day? If so, again I'd be tempted to do the same throughout.

*Thank you for the suggestion. The inclusion of extra data (next day 6 am) was only done in the rare cases when there were major gaps in the usable satellite track coverage of the day. In this case, although the method applies a weighted mean of all the data with its estimated error variance, there is little overlap (and none in the 'gap') and thus the averaging does not 'reduce' the error of the product. We could have included the 6pm data from the previous day, and done this systematically, but then the period representative of the product starts to be much longer than the roughly 12-16 hours of the data usually included for a specific day, and we lose some of the snapshot vision that the product provides. We believe that this 'time-smoothing/filtering' will be better done by future dedicated products that are currently developed at CATDS.*

l203-4 Brief comparison stats could be included here, eg bias and RMSE.

*Thank you for the comment. It is indeed interesting, in our study however only the flux computed from the ERA5 windspeed is use quantitatively (comparison between the reconstructed flux and Landschützer 2020 climatology, both computed from ERA5 windspeed). The process study based on the ship data is done using fCO2, so the ship winds will not influence the results of our study. We now precise it in the manuscript.*

l220-1 This sentence doesn't make sense - my guess as to your meaning would be something like 'Chl-a is hard to distinguish from terrigenous detrital material using ocean colour where both are present as they have similar spectral effects' or similar. Phytoplankton produce their own detritus, the effect of which is included in satellite chl-a algorithms.

*Indeed, thank you for the better phrasing.*

l235 ...prevents oscillations...

*Done*

l237 ...is ~4 uatm (or is 4 uatm if the 'of' was just a typo)

*Done (its ~4uatm)*

l248 ...over... It might be interesting to check the sensitivity to these extrapolations by calculating the mean fCO2 and/or flux without the extra points, equivalent to setting pixels outside the in situ range to the mean so they don't affect the result.

*Thank you for the suggestion, it was done in the first place, and the only effect is too fill gaps.*

l255 Rather vague and irreproducible with different data as it stands. Did you have a threshold of coverage? Given that in the end you average over all days, why do you not average all valid fCO2 values in a given pixel, regardless of coverage? If there's a specific reason (e.g. strong, consistent temporal gradients of gap location, which could bias the results), please state it clearly along with your exclusion criterion. Alternatively, how about doing it both ways, the difference suggesting a lower limit to uncertainty?

*It is indeed an interesting suggestion, that can work for the cloud gaps in chla and SST. However, the main problem is the absence of salinity data, and in this study, the threshold was based on salinity (no track in the area covering the plume). With the new product in development at CATDS, missing SSS data won't be a problem anymore and we agree that we should compute $fCO_2$ for every pixel, regardless of coverage in SST and CHLA.*

l284 I agree with Referee #3. This amounts to a 3D classification of your data, and there are many ways to achieve this. How did you do it? Manually? How did you arrive at 6 classes? What are the thresholds? Some are scattered through the text, but not in sufficient detail for me to be able to uniquely assign a (S,T,C) triplet to a class (or none). A simple table or a decision tree would suffice to make them reproducible. Do you have any interpretation at all of the grey data, which constitute a large proportion of the warmer waters?

*Indeed, we identified the 6 water masses manually, and defined thresholds. A table has been added with the different criteria for each water masses. Our interpretation for the grey data is mainly mixed water between several water masses. Some grey data (near Guadeloupe for example) might belong to another water mass, but in this study, we don't analyse the near-island water properties (mainly because satellite SSS might be doubtful there).*

|  | NASW | NBC | Modified NBC | Freshplume | Shelf | Filament |
|---|---|---|---|---|---|---|
| Temperature (°C) | <27.2 | > 27 | 27.16 < SST < 27.6 |  | < 26.6 | < 27.4 |
| Salinity | 35 < SSS < 36 | > 36 | > 35.6 | < 34.5 |  | 35.8 < SSS < 36.3 |
| Chlorophyll-a (mg.m$^{-3}$) | < 0.14 | < 0.14 | 0.11 < Chl$a$ < 0.25 | > 0.25 | > 0.25 | > 0.25 |

**Table 1.** Thresholds in SSS, SST and Chl$a$ used to define the 6 water masses identified.

l297 You should refer to this as simply 'NBC' rather than surface-intensified. It took me awhile to work out that this wasn't 'modified NBC' with a different name. Or rephrase along the lines of 'The NBC, intensified at the surface,...'

*Thank you, we rephrased.*

l334 was it likely to be affected by topography in this area?

*Yes, indeed, we reckon that its stationary position is driven by the topography as well.*

l345 ...processes are...

*done*

l355 SSS<34.5 appears to be the only necessary criterion in Fig 4b - is the chla limit a threshold or an observation?

*It is an observation, so we removed the chla value as it is not a threshold.*

l366-7 please quote the background silicate for comparison

*done*

l371 ...and modifies...

*done*

l380 Are you confident it's both? I don't think you have any independent measurements. If not, I suggest 'and/or'.

*done*

l386 ...A2's westward...

*done*

l387 from Fig 4b it looks like SST goes down to about 26.8C. Which of these are thresholds and which observations?

*All thresholds are now mentioned in a table, so the text is observations.*

l399 ...CO2 flux maps... Are these calculated from mean dfCO2 and everything else daily?

*We first computed CO2 flux maps for every available day (so that eventual correlation between dfCO2 and exchange coefficients are taken into account), and then averaged over the month of February. The only parameter constant is the atmospheric CO2 concentration, taken as the value in Barbados for February 2020, while other parameters are at a daily resolution.*

l400 Oddly vague - what is the resolution?

*The resolution is a blend of the resolution of the 3 satellites product used, so on the order of 60km for salinity, whereas data are reported at higher resolution (~2.5 km) for SST and Chla. We expect the fCO2 product to have therefore a resolution close to 2.5 km, but changing depending on the dominant relation (fCO2 to T,S or Chla). We added this precision to the manuscript.*

l407 (and abstract) personally, I consider /month to be a mangled unit, especially since your data comprise a subset of February! Why not quote /day, valid over the day range used for the calculation?

*Thank you for this interesting comment, we chose this unit to facilitate the comparison with other studies.*

l411 Not Feb but Feb 2-19

*Done*

l420 ditto

*Done*

l424 The two NBC...positive air-sea CO2 flux average in early to mid-February.

*Done*

l427-8 ...over 18 days

*Done*

l430 ...on average in early to mid-February...

*Done*

l441 ...in early to mid-February

*Done*

l456-7 ...associated with...

*Done*

l459 Is this how you did it? I thought it was T, S and C thresholds!

*Indeed you are right, it is through T,S and C thresholds. The boundaries are time varying and we removed the word "geographical" that could confuse the reader.*

l464-8 It would be easy to calculate overlaps between your water masses and the relevant Longhurst provinces, e.g. 90% of the pixels we classify as NASW are in the NATR province.

*Thank you for this interesting suggestion. This part has been modified and included in part 3.2, following the suggestions of the referee#3, and is now a bit less at the heart of our study. It would be nevertheless interesting, but a bit challenging since the NATR boundaries are quite vague.*

l471 ...presents...of air-sea CO2...

*Done*

l472 ...and in February 2020 we estimate the 5-16N, 59-50W domain to be a...

*Done*

l473 ...large-scale...

*Done*

l475 ...considerably smaller in...

*Done*

l476-8 can you quantify this?

*The new figure added gives a qualitative overview of the interannual variability of the fC02 in the region to address this issue.*

l481 ...and is...

*Done*

l491 ...current...

*Done*

l496 ...due both...

*Done*

l551 ...filaments...

*Done*

l554 ...plume, water...

*Done*

l558 ...regimes...

*Done*

l560 ...in early to mid-February...

*Done*

l561 You quote 10 times smaller earlier, which is it?

*Ten, thank you for noting this mistake.*

l561 ...contributes most... or ...is responsible for...

*Done*

l564 …contributes almost… or …is responsible for almost…

*Done*

l574 …than those of temperature and…

*Done*

l588 …, sampled daily on a 25 km x 25 km grid,…

*Done*

---

## Editor Decision (ED1)

**Editor's Comments on Authors' Response to RC1: Referee #3**

**Title:**
Authors may like to consider following title for the manuscript:

"Impact of North Brazil Current rings on air-sea CO2 flux variability in western tropical Atlantic: Evidence for larger CO2 uptake during wintertime"

**Abstract:**
Suggested changes:

The key processes driving the air-sea CO2 fluxes in the western tropical Atlantic (WTA) in winter are poorly known.  WTA is a highly dynamic oceanic region,  expected to have  dominant role  on the variability of CO2 air-sea flux. In early 2020 (February), this region was the site of a large in situ survey  and studied in  wider context through satellite measurements. In  the winter (February), the North Brazil Current (NBC) flows northward along the coast of south America, retroflects close to 8°N and pinches off the world's largest eddies, the NBC rings. The rings are formed to the north of the Amazon River mouth when freshwater  discharge is still significant in winter (a time period of relatively low runoff). We show that in February 2020, the region [50°W-59°W – 5°N-16°N] is a CO2 sink from the atmosphere to the ocean (-1.7 TgC.month-1), a factor of 10 greater than previously estimated. The spatial distribution of CO2 fugacity is strongly influenced by  eddies south of 12°N. During the campaign, nutrient rich freshwater plume from the Amazon River is entrained by a ring from the shelf up to 12°N leading to high phytoplankton concentration and  significant carbon drawdown (~20 % of the total sink).  In trapping equatorial waters,  NBC rings  are a small source of CO2. The less variable North Atlantic subtropical water extends from 12°N northward and represents ~60 % of the total sink due to their lower temperature associated with winter cooling and strong winds. Our results, in identifying the key processes influencing the air-sea CO2 flux in the WTA, highlight the role of eddy interactions with the Amazon River plume. It sheds light on how  lack of data impeded a correct assessment of the flux in the past, and on the necessity of taking into account features at meso and small scale.